# How Reasoning Evolves from Post-Training Data in Sequential Decision-Making Domains

## Abstract

We study how reasoning evolves in a language model – from supervised fine-tuning (SFT) to reinforcement learning (RL) – by analyzing how a set of theoretically-inspired datasets impacts language model performance in chess. We find that fine-tuning a model to directly predict the best move leads to effective RL and the strongest downstream performance – however, the RL stage elicits *unfaithful* reasoning (reasoning inconsistent with the chosen move). Alternatively, training on multi-move trajectories yields comparable downstream performance with faithful reasoning and more stable RL. We show that RL induces a substantial positive shift in the distribution of move quality and reduces hallucination rates as a side effect. Finally, we find several SFT-checkpoint metrics – metrics spanning evaluation performance, hallucination rates, and reasoning quality – to be predictive of post-RL model performance. We release checkpoints and final models as well as training data, evaluations, and code which allowed us to surpass leading open-source reasoning models in chess with a 7B-parameter model[1].

## 1 Introduction

*What is required to train a language model to reason through RL?* Several ingredients appear critical – a strong base model and a compatible domain are sensible starting points. *But what is a strong base model? And once you have a domain, how do you train it to reason effectively?*

We seek to address these motivating questions by training a language model to reason in a verifiable, sequential decision process. Specifically, we choose chess as our focus because of several convenient elements: intrinsic difficulty for LLMs, established theory, favorable structure (episodic MDP), large datasets, and an efficient oracle (chess engines) for verifiable rewards and high-quality synthetic data generation. As a result, we can measure how different training data influences our language model through SFT and how RL further evolves reasoning from this checkpoint in a controlled setting.

Language-based reasoning (OpenAI, 2024) has emerged as a promising technique to advance language model capabilities, although much research remains confined to specific domains such as math and coding. Reasoning, often characterized as extending a model's "chain-of-thought" behavior (Wei et al., 2022) using methods such as RL, benefits from these domains being clearly verifiable: the math is correct or the code passes all tests. This verifiable nature – combined with downstream applicability and skill transfer – has stimulated research in these specific domains. As a result, reasoning models have achieved profound results across many related benchmarks (OpenAI, 2025a; Anthropic, 2025; Google DeepMind, 2025), long-duration tasks (Kwa et al., 2025; METR, 2025), and even earned an International Math Olympiad gold medal (DeepMind, 2025; OpenAI, 2025).

Although recent work explores reasoning in more subjective domains (Whitehouse et al., 2025) through techniques such as "LLM-as-a-judge" (Zheng et al., 2023), we focus here on those that are verifiable. The core verifiable domains – math and coding – benefit from years of continued research that has established a corpus of high-quality training data which produces strong out-of-the-box base model performance. While we benefit from stronger general base models, working on these explored domains would pose a challenge in isolating the influence of our training interventions.

---

[1]Code and models available at lang-chess.

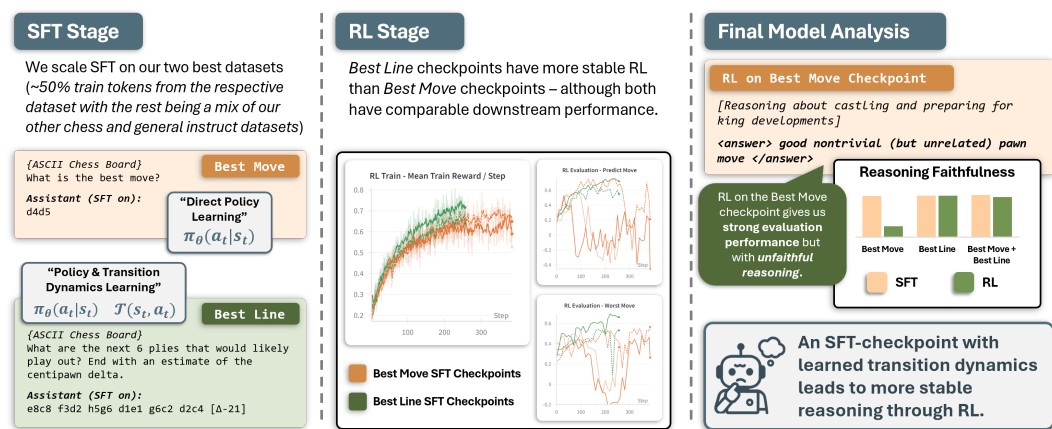

Figure 1: Following initial data inclusion experiments, we scaled SFT on our two best-performing datasets. Both resulted in comparably strong final evaluation performance, but training on optimal move trajectories (*Best Line*) led to more stable RL and faithful reasoning compared to training on single best moves (*Best Move*).

Chess thus emerges as an attractive domain for studying reasoning. Language models have historically struggled with chess (Acher, 2023; Dynomight, 2024) and even state-of-the-art reasoning models still falter – often responding with illegal moves or simple blunders as seen in the August 2025 Kaggle AI Chess Exhibition (Kaggle & Google DeepMind, 2025). Models underperform partly because chess data, while abundant, is rarely emphasized in pretraining; further, the game's combinatorial structure makes generalization difficult. While this poses a difficulty, access to superhuman verification (in chess engines) provides an efficient method for verifiable rewards and synthetic data generation. All these features make chess a compelling testbed for reasoning.

Motivated by this setting, we train a 7B-parameter model on custom datasets using both SFT and RL to achieve performance surpassing gpt-oss-120b (OpenAI, 2025b) on several benchmarks. Our study centers on the following questions which we will address:

- *Q1*: How do different datasets (*e.g., programmatically generated, synthetic rejection sampling, synthetic from a harness*) impact downstream performance after SFT and RL?

- *Q2*: How does RL influence a model's qualitative behaviors (*e.g., move quality distribution, reasoning strategies used, rate of hallucination*)?

- *Q3*: Which SFT-checkpoint metrics are predictive of final RL performance?

We show that focused SFT on predicting a single best move (*Best Move*) leads to strong performance but *unfaithful* reasoning through RL; on the contrary, training on multi-step move trajectories (*Best Line*) has more stable RL and faithful reasoning. We find that RL leads to fewer hallucinations and a substantial positive shift in move quality, and we see that several SFT-checkpoint metrics (both *qualitative* and *quantitative*) are predictive of final RL performance.

## 2 RELATED WORK

### 2.1 REASONING IN LANGUAGE MODELS

*Reasoning* in causal language models can be interpreted as self-guided search that makes a task more tractable. Consider a numerical math problem: effective reasoning should increase the probability of producing the correct number *more* than if the model had immediately predicted the final answer. Note that this reasoning need not be wholly interpretable – for example, it can exist in continuous space (Hao et al., 2024) or shift between languages (DeepSeek-AI et al., 2025) – what ultimately matters is that the intermediate steps are beneficial to the model. For this work we will focus on language-based reasoning.

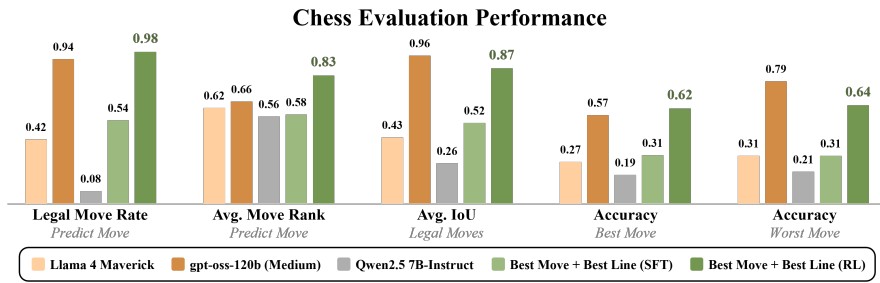

Figure 2: Performance of our best reasoning model trained from a Qwen2.5 7B-Instruct base across our evaluations. Note that trivial performance (i.e., random guessing) is 0.2 for the *Best Move* and *Worst Move* tasks. See Appendix B for example evaluation questions.

The era of *reasoning models*, notably initiated with OpenAI's release of o1 (OpenAI, 2024), builds upon much prior work in self-guided in-context adaptation. Models, when told to work "step by step" and write down intermediate results on a "scratchpad" (Nye et al., 2021), saw performance improvement on multi-step computations – this result was reinforced at scale and termed "chain-of-thought" in later work (Wei et al., 2022). Further, these reasoning traces can be used for iterated improvement through fine-tuning on successful generations as evidenced by STaR (Zelikman et al., 2022). Quiet-STaR (Zelikman et al., 2024) extended this from fine-tuning by using a reinforcement learning policy-gradient update in REINFORCE (Williams, 1992) over tokens influenced by intermediate reasoning steps. This iterated bootstrapping using reinforcement learning for policy-gradient updates has been the primary underlying method fueling the latest developments in reasoning models.

Following OpenAI's release of o1, many leading systems began incorporating similar reasoning techniques to improve performance. Notably, DeepSeek-R1 (DeepSeek-AI et al., 2025) and Kimi k1.5 (Kimi Team et al., 2025) were among the first reasoning models to effectively approach state-of-the-art ability and publicize the underlying training methods.

## 2.2 Reasoning through RL

While there exist several effective methods for training models to reason such as in-context prompting (Wei et al., 2022; Kojima et al., 2023), model distillation (DeepSeek-AI et al., 2025), or SFT on successful outputs (Zelikman et al., 2022; Yuan et al., 2023), we focus our attention on the setting of applying RL to improve model reasoning.

RL has been used as an effective tool to guide model behavior with Ouyang et al. (2022) inciting the viral *ChatGPT moment* that brought language models to public attention. Successful RL – regardless of the setting – requires valuable reward signals; for language models these rewards can be generated using the following methods: rewards can be parsed and automatically calculated in verifiable tasks (Shao et al., 2024), determined directly through human judgment (Christiano et al., 2017), scored with a learned reward model (Ouyang et al., 2022), or elicited using a language model as a judge (Whitehouse et al., 2025). Rewards can be generated for the entire outcome or at intermediate steps (Lightman et al., 2023), and learned value functions can approximate credit-assignment at the token-level (Schulman et al., 2017) or reward can be indiscriminately applied over a full sequence (Shao et al., 2024). While this covers many methods in RL for language models, it is not exhaustive.

A common family of algorithms used in RL for language models is Proximal Policy Optimization (PPO) (Schulman et al., 2017) – an actor-critic method. Because actor-critic methods require learning a value function to address the credit-assignment problem (which can be computationally expensive and experience instability), new methods such as Group Relative Policy Optimization (GRPO) (Shao et al., 2024; DeepSeek-AI et al., 2025) have emerged to remove this learned value function requirement. GRPO has further evolved through variants such as Dr. GRPO (Liu et al., 2025), which removes sequence-level length normalization, and DAPO (Yu et al., 2025), which removes the KL penalty, increases the clipping bound to encourage exploration, and addresses length normalization issues observed in GRPO.

## 2.3 CHESS ENGINES

Computer scientists have developed grandmaster-level chess systems built on three notable techniques: 1) Classical search-based engines such as IBM Deep Blue (Campbell et al., 2002) or Stockfish that use a minimax-based search algorithm (commonly alpha-beta pruning), 2) neural search-based systems such as AlphaZero (Silver et al., 2017) and its open-source implementation in Leela Chess Zero that learn policy functions through RL self-play combined with Monte Carlo Tree Search, and 3) searchless neural systems such as Google DeepMind's chess transformer that predicts a move directly from a board state (Ruoss et al., 2024). The respective 40/15 Elo scores of Stockfish and Leela Chess Zero as of August 8, 2025 are 3645 and 3444 (Computer Chess Rating Lists, 2025), and DeepMind's chess transformer reached a Lichess blitz Elo of 2895 (Ruoss et al., 2024). While it is worth noting that recent versions of Stockfish use neural networks to estimate the value of board states – it still largely employs the same core algorithm used by classical search-based engines.

As discussed previously, language models struggle in the domain of chess. However, it is worth mentioning gpt-3.5-turbo-instruct which has an estimated Elo around 1700 (Acher, 2023). While this anomaly is interesting, this performance isn't from language-based reasoning – rather it is direct next move prediction (i.e., only outputs the move to play). Google DeepMind's chess transformer validated that a 270 million parameter transformer is capable of reaching grandmaster-level chess without search (*this was achieved by learning a value function, though a policy function was also tested*). As far as the authors are aware, no language model has achieved competitive-strength chess ability through language-guided reasoning and the best reasoning model in chess is OpenAI o3 which won the 2025 Kaggle AI Chess Exhibition (Kaggle & Google DeepMind, 2025).

## 3 BACKGROUND

Our analysis is focused on the Qwen2.5 7B-Instruct model (Qwen et al., 2025). Given the baseline model has insufficient ability, we first conducted SFT prior to the RL stage. We began with a full set of data inclusion studies – from SFT to RL – to determine the most effective recipe before doing a final, scaled training run on our leading mix.

### 3.1 BOARD AND MOVE REPRESENTATION

For all training and evaluation we provide the board state in a visual ASCII-format. We ran preliminary tests on several board formats including Forsyth-Edwards Notation (FEN), FEN with space delimiters, and a visual ASCII-format. While these showed similar quantitative performance, we opted for the visual format following subjective qualitative analysis. Appendix A provides examples of the considered board states and discusses tokenization limitations in each. Note that our board representation omits move repetitions due to dataset limitations – in competition chess, repetitions can be used as a termination condition. However, since none of our evaluations incorporates repetitions, we can view our representation as a Markov-complete state.

For move representation, we follow DeepMind's chess transformer (Ruoss et al., 2024) and represent all moves in Universal Chess Interface (UCI) format (e.g., `e1e2`). This decision was made in lieu of formats such as Standard Algebraic Notation (SAN) which may be more commonly represented in training data – SAN has intricacies that could evoke errors avoidable by using UCI notation.

### 3.2 EVALUATIONS AND RL ENVIRONMENT

We created four custom tasks that we use for evaluations and the RL training environment. For the **Predict Move** task, we provide a board and ask the model to play the best move – no list of legal moves is provided. We measure both the ratio of legal moves generated as well as the move quality for legal moves provided. Move quality is measured as the normalized rank among legal moves ($\in [0, 1]$) as determined by a chess engine – where the best move is given a score of 1 and the worst move a score of 0. For the **Best Move** and **Worst Move** tasks, we provide a board and a set of 5 moves. The task is to choose the best move (and worst move, respectively) of the candidate moves provided. For both tasks, candidate moves are sampled such that there is a threshold of move quality (determined by a chess engine) separating the correct answer from other candidates. Finally, the **Legal Moves** task asks the model to, for a given board and piece, list out all the legal moves that

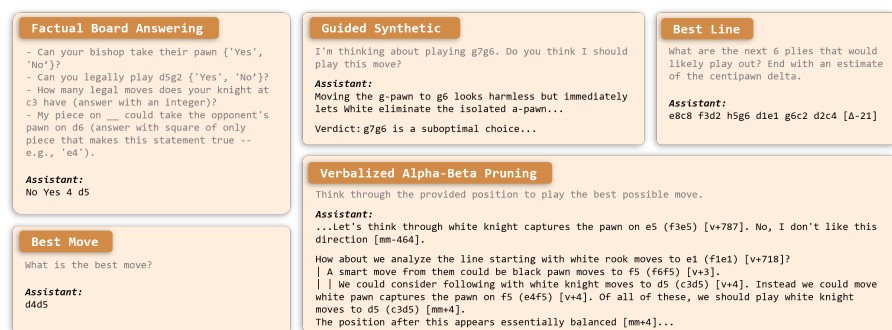

Figure 3: Samples from a selection of the custom datasets. The gray font represents an abbreviation of the core prompt – in all samples the model is trained with a verbose instructive prompt and provided with a board in our visual ASCII-format. Full samples are included in Appendix C.

piece can make. Results are computed as intersection over union (IoU) versus the ground truth. We provide example questions in Appendix B.

## 3.3 DATASETS

We created several theoretically-inspired datasets to study training dynamics from SFT to RL. Consider that chess can be represented as an MDP. At time $t$ there is a state $s_t \in \mathcal{S}$ and playing a ply (i.e., half-move) from the player's perspective constitutes an action $a_t \in \mathcal{A}(s_t)$. We abstract the opponent's move to the environment, viewing a state transition as $s_{t+1} \sim \mathcal{T}(s_t, a_t)$. Additionally, for each board state-action pair there is a reward $r_t = \mathcal{R}(s_t, a_t)$ which we can approximate using a shaped dense reward (centipawn delta, i.e., the change in an engine's board evaluation measured in hundredths of a pawn) from a chess engine: $r_t = \gamma V_{\text{engine}}(s_{t+1}) - V_{\text{engine}}(s_t)$ with $\gamma = 1$. We will use this formulation to discuss motivation for several of our custom datasets.

We provide a brief description of each dataset and will further elaborate on data design and motivation within the context of experimental results in Section 4. We include detailed explanations of each dataset and full examples in Appendix C – abbreviated examples are included in Figure 3. Regarding our datasets, we organize them into the following four categories:

- **General Instruction Following**: Specifically, Magpie Llama 3.3 70B (Xu et al., 2024).

- **Rejection Sampling**: We generate outputs from Llama 4 Maverick (Meta AI, 2025) on our four evaluation types. We chose Llama 4 Maverick for qualitative and quantitative performance, retaining samples from the *Best Move* and *Worst Move* evaluations if correct and keeping outputs from the *Legal Moves* and *Predict Move* evaluations if above a threshold.

- **Guided Synthetic**: We prompt Llama 4 Maverick and gpt-oss-120b with a programmatically generated harness. Specifically, we provide a beginning board, 5 plies (the first ply being a move candidate and following plies being optimal play from a chess engine), and the ending board state. The task is to generate an explanation of how the proposed candidate move will play out, ending with a final verdict for the proposed move.

- **Programmatically Generated Data**:

  - **Factual Board Answering**: We build on top of a chess engine to generate simple question-answer (QA) pairs for a given board. These questions may ask if a move is legal, which square is threatening a specific piece, or how many legal moves a piece has. We combine multiple QA pairs for each sample.

  - **Verbalized Alpha-Beta Pruning**: We use a custom program built upon Stockfish to sample moves, rollout the line of play for each move (with branching and board values), and verbalize rollouts and minimax decisions in natural language. We explicitly build in tree search reasoning strategies and sample poor moves to verbalize the process of *pruning*, and we leverage a large, custom prompt bank to add diversity to natural language outputs.

  - **Best Move**: Given a board, immediately predict the best move in UCI notation.

– **Best Line**: Given a board, predict the optimal line of play $(4 - 6$ plies) ending with the expected centipawn delta from playing this line.

### 3.4 TRAINING ENVIRONMENT

All SFT is conducted using LlamaFactory (Zheng et al., 2024) and all RL is conducted using veRL (Sheng et al., 2025). We utilize Dr. GRPO (Liu et al., 2025) for our RL optimization algorithm and employ the *Clip-Higher* strategy with no KL divergence per Yu et al. (2025). A full list of hyperparameters for both SFT and RL are included in Appendix H.

## 4 KEY FINDINGS

We ran a series of inclusion analyses to understand the efficacy of each data type and scaled our best-performing recipes. Figure 2 highlights the performance of our best reasoning model. We found the *Best Move* and *Best Line* datasets to be most effective – especially when lightly supplemented with our other, less effective datasets. Our scaled runs build off of the *Best Move - All* and *Best Line - All* datasets that use this dataset diversity. The best final performance was achieved by first training Qwen2.5 7B-Instruct on 60 million tokens (*Best Move - All* data) followed with 60 million tokens (*Best Line - All* data). Appendix D provides further detail on our experiments.

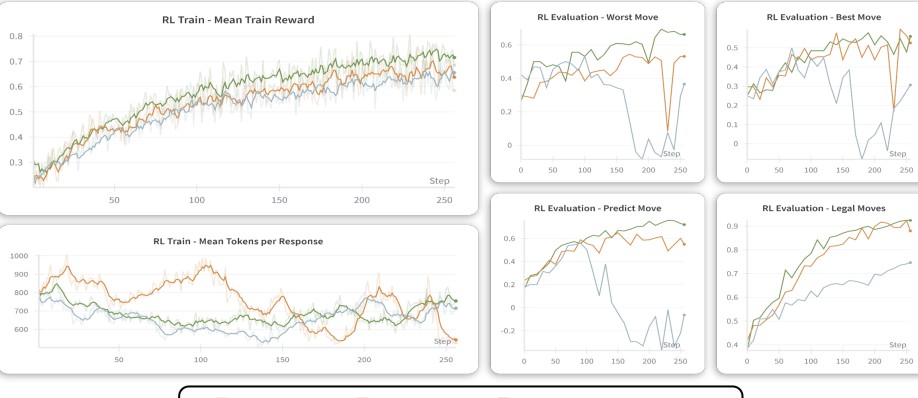

Figure 4: RL training performance on our scaled SFT-checkpoints. *Left*: Train reward and tokens per response (smoothed using an exponential moving average with decay factor 0.9). *Right*: Reward on the held-out evaluation set during training. The *Best Move* dataset, while having strong ending performance, experienced more unstable RL compared to scaled runs trained on *Best Line* data.

### 4.1 Q1: HOW DO DIFFERENT DATASETS IMPACT DOWNSTREAM PERFORMANCE AFTER SFT AND RL?

**Multitask training is beneficial** for a fixed token budget, yielding higher move quality, less reward hacking, and a generally more robust model. This is shown by a comparison between *Rejection Sampling (Predict Move)* and *Rejection Sampling (All Evals)*: for the former we SFT and conduct RL on only the *Predict Move* task – for the latter we use all tasks. Our experimental results (Figure 5) led to these takeaways. We incorporate multitask training in all following experiments.

**The most effective datasets were dense with difficult, high-quality tokens**. Consider the *Factual Board Answering* dataset: we designed this dataset to sample multiple QA pairs from a set of custom generators built atop a chess engine. The motivation is to force the model to embed complex board understanding in its latent layers through these immediate (often 1 token) responses; however, we find this dataset's performance comparable to our initial *Rejection Sampling (All Evals)* experiment. That is, latent board understanding did not result in a better reasoning model. If we compare this with a similarly dense task in *Best Move* (predict the best move directly) or *Best Line* (generate the optimal line of play), we see that performance is better when training on the latter datasets (Figure

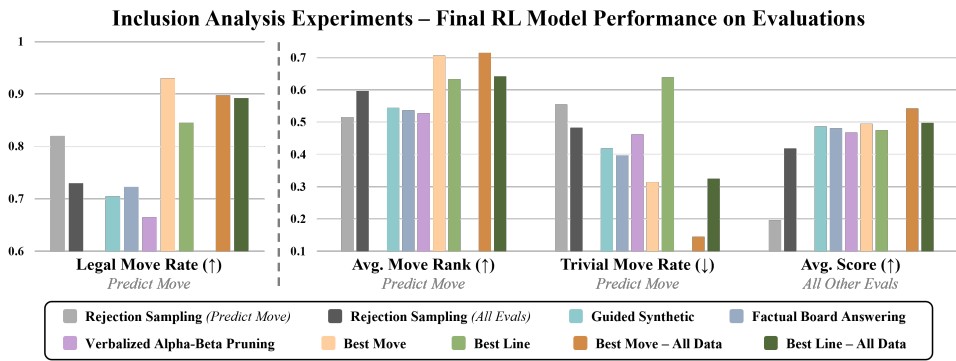

Figure 5: Evaluation results for final RL models from each data inclusion experiment. Within each metric we split results into three sections: *(left)* compares single vs. multitask, *(middle)* compares the targeted data inclusion experiments, *(right)* covers our data diversity experiments. Note that in all experiments we SFT on 15 million tokens and do RL on 8k samples. In the single task setting, RL only uses the *Predict Move* task; in all other settings the 8k samples are split evenly between our four task types. See Appendix D for detailed results (including exact token distributions).

5). Intuitively, these more difficult tasks require the model to develop a richer latent understanding (e.g., playing the best move inherently requires board understanding).

**Dataset diversity remains valuable**. We see that the *All Data* experiments for *Best Line* and *Best Move* are superior to their focused counterparts (Figure 5). For these runs, we SFT on nearly all our data types – this comes despite mixed results on several of the individual inclusion analyses. Notably, the *Verbalized Alpha-Beta Pruning* experiment showed it was detrimental for training – we created this dataset as it is hallucination-free and includes rollouts and value functions (instilling $V(s_t)$ and $\mathcal{T}(s_t, a_t)$). However, it possesses a lower density of high-quality tokens (moves and valuations) and is surrounded by memorizable prompts. The *Guided Synthetic* dataset produced subpar performance as well when compared against a *Rejection Sampling* baseline. Regardless, limited inclusion of all data was found to be beneficial.

***Best Line* had more stable RL training than *Best Move***. Figure 4 outlines RL training performance for our scaled runs – the models fine-tuned on *Best Line* data had more stable training dynamics. One reason may be that training on *Best Line* data – which includes multiple moves and ends with a valuation – allows the model to learn a world model for chess (both a value function $V(s_t)$ and transition dynamics $\mathcal{T}(s_t, a_t)$). This is further supported by the coming discussion on *reasoning faithfulness* that reinforces this stability observation. Additionally, we find that models trained on *Best Line* perform better than our *Best Move* experiments on evaluations not trained on during RL – validating our observation that training on *Best Line* data leads to a more robust policy. Detail on the out-of-distribution evaluation is included in Appendix D.

### 4.2 Q2: HOW DOES RL INFLUENCE A MODEL'S QUALITATIVE BEHAVIORS?

**Multi-step trajectory data led to the most faithful reasoners**. Note that we measure faithfulness with gpt-oss-120b judging alignment of final answers with reasoning traces. Our most faithful reasoner was achieved by training on the *Best Line* dataset which incorporates a rollout (in UCI) followed by a valuation (as a centipawn delta). This structure can be viewed as approximating $n$-step bootstrapping with $n = 2$ or 3 depending on ply depth. We can contrast this with the *Best Move* dataset which approximates a direct policy function (i.e., learning $\pi_\theta(a_t|s_t)$ via behavior cloning). Our multi-step trajectory checkpoints largely retained faithful reasoning through RL – on the other hand, the *Best Move* dataset became an *unfaithful* reasoner through RL, often displaying final, non-trivial answers that were disconnected from its reasoning trace. Appendix G provides further detail on the reasoning quality measurement and includes an unfaithful reasoning example.

This is interesting as the unfaithful reasoner improves through RL without defaulting to trivial moves. Further, this improved ability is not explained by longer generations (Figure 4). One possible explanation we offer is the following: faithful reasoning from multi-step data may arise due to the model

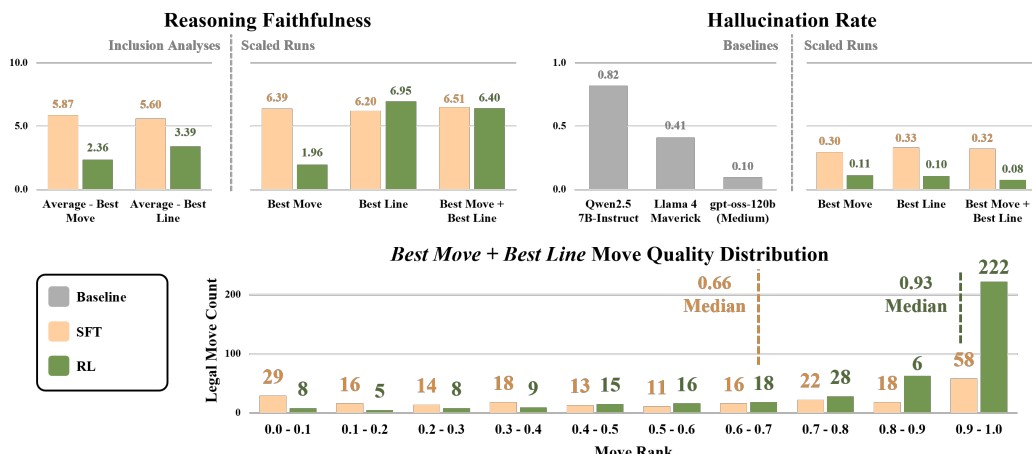

Figure 6: Key results highlighting how RL influenced our SFT-checkpoints. *Top left*: RL on the *Best Move* SFT-checkpoint induced *unfaithful* reasoning whereas checkpoints trained on multi-step data were more robust. *Top right*: RL drove a meaningful decrease in hallucination rate as a side effect of simply maximizing reward on our evaluations. *Bottom*: Our *Best Move + Best Line* scaled run saw a significant distribution shift after RL in its move quality on the *Predict Move* evaluation ($n = 400$). This shift is an improvement on both an absolute and relative basis, highlighting the efficacy of RL. Appendix G has further detail on our reasoning quality measurement and Appendix E outlines hallucinations.

internalizing a chess world model (transition and value functions), whereas unfaithful reasoning may result from strong latent capability mixed with weak verbalized reasoning ability. Previous work has found that models (Turpin et al., 2023) may attempt to rationalize their answers in chain-of-thought unfaithfully if they are biased; in our case, the model may be attempting to rationalize the move it has "already decided".

Regardless of reasoning faithfulness, **RL drove a substantial positive shift in move quality played** (Figure 6). Not only does RL improve the frequency of the best moves being played but it also decreased the frequency of low quality moves on an absolute and relative basis. Additionally, **RL reduces the rate of hallucinations within reasoning traces** (Figure 6). This result is a side effect of rewarding correct answers as we do not incentivize factuality – we provide further detail on hallucinations in Appendix E and show that this result is shared across all data inclusion experiments.

Lastly, we analyzed reasoning strategy usage at both the SFT and RL model stages. This follows prior work (Gandhi et al., 2025; Zeng et al., 2025) showing that effective reasoning models tended to utilize more reasoning strategies. We did not see clear trends in our analysis apart from our weaker models – specifically those more prone to reward hacking – almost exclusively reducing the usage of reasoning strategies through RL. In contrast, stronger models had mixed usage trends. We defer to Appendix F for further detail.

### 4.3 Q3: WHICH SFT-CHECKPOINT METRICS ARE PREDICTIVE OF FINAL RL PERFORMANCE?

Finally, we conducted a simple linear regression analysis comparing metrics from the SFT-checkpoint with the final RL model's performance (average over all evaluations). Figure 7 highlights three SFT-checkpoint metrics that are statistically significant predictors of downstream performance.

Some of this is expected – an SFT model that scores higher on evaluations is likely better suited for RL. However, we find that more qualitative signals (specifically, referenced move accuracy and reasoning quality) are also predictive of downstream performance. This shows that an effective SFT-checkpoint is one that is truthful (low hallucination rate), already an effective reasoner, and exhibits strong performance in the domain of focus.

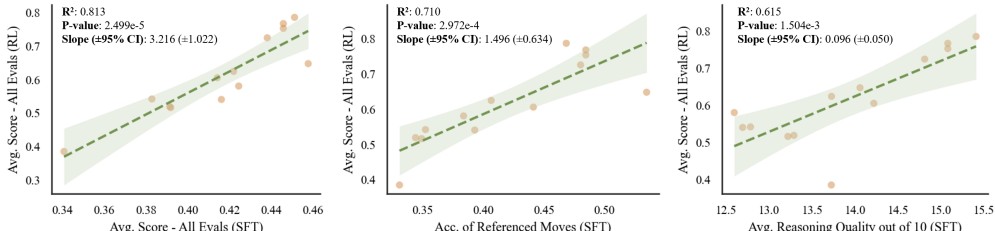

Figure 7: Linear regression comparing the final RL model (average score over all evaluations) with various metrics from its corresponding SFT-checkpoint. *Left*: Vs. average score over all evaluations. *Middle*: Vs. percent of moves referenced during reasoning trace that are legal (parsed by Llama 4 Maverick). *Right*: Vs. reasoning quality (mean over all reasoning quality metrics as judged by gpt-oss-120b). Shaded region represents the $95\%$ confidence intervals.

## 5 LIMITATIONS & FURTHER DISCUSSION

To begin, the intention of this work has always been to study general reasoning properties in language models. Thus, while the final evaluation of our model is a welcome result, we focused much of our effort on understanding the qualities and development of reasoning; this means that there are many methods we believe could further improve a chess reasoning model beyond our final RL model. For example, in full-game play our final RL model had poor performance against OpenAI o3. We suspect some level of distribution mismatch: training emphasized mid- and late-game positions to reduce trivial moves, which likely degraded opening play and hurt head-to-head results versus an opponent with stronger opening theory.

We have also identified several unexplored techniques that could increase performance in our final RL model. We minimally experimented with reward function tuning in our RL environment and expect focused effort could improve performance – particularly on the *Predict Move* task. Further, incorporating multi-turn RL and chess puzzles to more closely mimic a full chess game would likely yield strong improvements.

Regarding model choice, we acknowledge that our experiments were confined to Qwen2.5 7B-Instruct – while it would have been valuable to replicate on distinct base models, due to constraints this was not pursued. Additionally, we chose gpt-oss-120b as our comparator because, in tests against Kimi K2 and DeepSeek-R1-0528, it showed state-of-the-art open-source performance and was more convenient to run with our available resources.

Finally, we believe that there is a promising direction in training on environment dynamics to improve general reasoning abilities that is beyond the scope of this paper. Recent works such as CodeI/O (Li et al., 2025) and Absolute Zero (Zhao et al., 2025) show that training a model to predict program outputs from both code and inputs produces strong reasoners. We believe this can be pursued in other sequential domains as well – for example, dialogue or embodied agentic tasks – where the model can be trained on both actions and responses.

## 6 CONCLUSION

We conduct a detailed study of how various custom datasets influence training dynamics through SFT and RL in the domain of chess. Our analysis highlights that training to predict the best move directly produces strong downstream performance but comes with *unfaithful* reasoning. Instead, training on multi-move trajectories delivers similar performance with faithful reasoning. We find that RL leads to fewer hallucinations and a substantial positive shift in move quality, and we see that several SFT-checkpoint metrics (both *qualitative* and *quantitative*) are predictive of final RL performance. We publish our code and data as well as scaled SFT-checkpoints and RL models.

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

# A  BOARD FORMAT

We tested various board representation formats – the three formats shown in Figure 8 had similar initial evaluation performance on a baseline Qwen2.5 model. However, upon qualitative analysis Visual (ASCII) format was ultimately chosen. Additional rationale and comments on each are listed below:

- **FEN**: The tokenizer combines specific characters (e.g., \n, RK, PPP) and this may limit generalization. Additionally, uneven tokenization across rows may hinder spatial understanding.
- **Spaced FEN**: While this format resolves combined character issues, there is an inconsistent representation of spaces – ' 2' is two tokens while ' p' is one token. This may present issues in downstream spatial understanding.
- **Visual (ASCII)**: Ultimately chosen because it alleviates concerns mentioned in Spaced FEN.

*Note: We recommend that future practitioners alter the Visual (ASCII) format. Qwen-series (2 and 3) and Llama-series (3 and 4) tokenizers treat ' . \n' as a single token with ' p\n' as two tokens – this can be fixed by including a space before each newline. This inconsistency was discovered late in training and thus not integrated into our project. We include an updated* uniform_visual *board format in our released code that improves upon Visual (ASCII).*

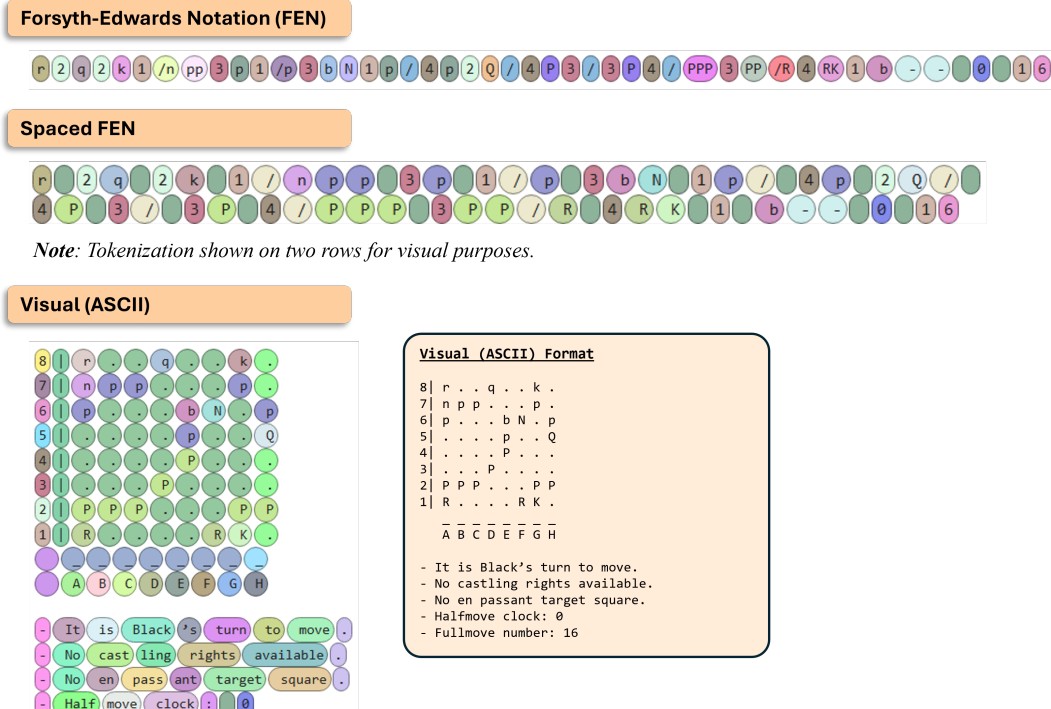

Figure 8: Visualized tokenization of three candidate board formats using the Qwen2.5 tokenizer.

## B    EVALUATION SAMPLES

Figure 9 contains an example of each evaluation type for the displayed board.

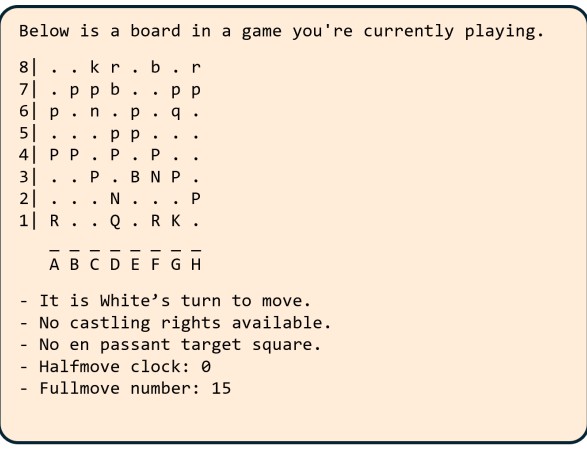
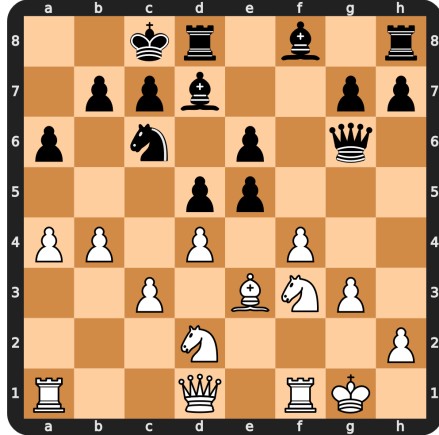

```
Below is a board in a game you're currently playing.

8| . . k r . b . r
7| . p p b . . p p
6| p . n . p . q .
5| . . . p p . . .
4| P P . P . P . .
3| . . P . B N P .
2| . . . N . . . P
1| R . . Q . R K .

   _ _ _ _ _ _ _ _
   A B C D E F G H

- It is White's turn to move.
- No castling rights available.
- No en passant target square.
- Halfmove clock: 0
- Fullmove number: 15
```

**Predict Move**

Q: "Determine the best move from this position and return it within answer tags."
A: *{'b4b5': 0.745, 'd4e5': 0.609, 'f4f5': ... }. get(parsed_answer, IllegalMoveError())*

**Best Move**

Q: "You must choose the best move from the following moves: ['a4a5', 'g1h1', 'd1b3', 'b4b5', 'a1a2']. "
A: *b4b5*

**Worst Move**

Q: "You must choose the worst move from the following moves: ['g1f2', 'a4a5', 'd1c2', 'd1b1', 'f4f5']. "
A: *d1c2*

**Legal Moves**

Q: "You must provide a list of all legal moves for the white rook at a1."
A: *[a1b1, a1c1, a1a2, a1a3]*

Figure 9: Example questions for each evaluation task on the same board. Note that in actual prompts (omitted in the figure) we include information related to the required format for valid parsing.

## C DATASET TYPES AND SAMPLES

We now outline more detail on the format and creation of our datasets.

For **General Text Alignment**, we use samples from the Magpie (Xu et al., 2024) Llama 3.3 70B dataset as a means of regularization. Given that a significant portion of the SFT data is dense in chess notation, we felt it valuable to maintain the instruction-following abilities of our trained model to prevent performance collapse.

Our **Rejection Sampling** data comprises of four separate tasks, `Best Move`, `Worst Move`, `Predict Move`, and `Legal Moves`. We chose to use Llama 4 Maverick to generate synthetic rejection-sampled data as it offered strong performance, was efficient to run on our hardware, and offered more natural-sounding reasoning compared to other reasoning models. Our motivation for this dataset was to instill correct reasoning format and bake in ability from the teacher model given Qwen2.5 7B-Instruct had incredibly poor base performance. This motivation follows Yuan et al. (2023) which found rejection sampling fine-tuning to be highly effective for improving weaker models.

For each evaluation set, we sampled outputs from Llama 4 Maverick (Meta AI, 2025) – retaining answers that were correct on `Best Move` and `Worst Move` and keeping outputs on `Predict Move` and `Legal Moves` if the answers met a threshold. This data was further sampled to ensure a desirable distribution regarding move count and piece types, as rejection sampling inherently shifts the distribution towards board states that the data-generating model performs well on.

```
Rejection Sampling (Predict Move) Sample:

Assistant (SFT):
<think>
First, let's analyze the given position:
The current state of the board is:
8| r n . q . b n r
7| p . . . p k . p
6| . p . . . p .
5| . . p . p . . .
4| . . . . . . . .
3| . . . P . . P .
2| P P P . . P . P
1| R N B . K . . .

It's Black's turn. We need to assess the position carefully.

The first observation is that Black's king is somewhat exposed on e7, and there's no clear indication that it's
going to castle soon since the kingside pieces haven't moved much. However, we also see that White's king is
still in the center and hasn't castled yet, although it has the right to castle queenside.

...

A straightforward plan involves moving the f8 rook into play or preparing to develop other pieces effectively.

Looking closely at possible moves, `g7g6` appears too passive without directly contributing to rapid
development or attacking play. Let's evaluate more active options like knight moves or pushing central pawns
when feasible.

However, upon closer inspection, I notice that `b8c6` develops a piece, supports potential central pushes
(`d7d5`), and doesn't overly expose Black. It's a solid developmental move.

Another consideration is assessing whether there are direct attacks or tactical shots available. For instance,
checking if any of White's under-developed status leaves them open to tactics.

Upon deeper review, I believe `b8c6` fits well because it develops a piece harmoniously and gets ready for
further actions such as controlling the center or supporting potential breaks.

</think>
<answer> b8c6 </answer>
```

Figure 10: A *Rejection Sampling* example. The full response in the sample is shortened for space. Note that a drawback of this dataset is that it is prone to hallucinations as is shown in the provided sample.

To construct our **Guided Synthetic** data, we generate synthetic data by using a sufficiently strong teacher model to verbalize outcomes of a move. A teacher model (Llama 4 Maverick or OpenAI gpt-oss-120b) is provided with a beginning board state, line of up to 5 total plies (where all plies following the first move are the top suggested chess-engine move), and an ending board state. The model is tasked with verbalizing the merit of the proposed ply – first providing logic then a verdict on the candidate move's quality (given how the board would develop).

In the MDP setting, this can be interpreted as verbalizing $n$-step bootstrapping (Sutton & Barto, 2018) with $n = 3$ (given 5 plies yields 3 player actions). This is due to a verbalized transition function in the logic (outlining how the board will develop – $\mathcal{T}(s_t, a_t)$ ) combined with a verbal-

ized value function $(V(s_t))$ in the verdict. The intention of the harness is multifaceted – ideally it should reduce hallucination rate while also guiding the model to generate this verbalized $n$-step bootstrapping format with meaningful rationales and verdicts.

```
Guided Synthetic Sample:

User:
Below is a board in a game I'm currently playing.

8| r . b . . r k .
7| . p . . b p p p
6| . q p p . n . .
5| . . . P p . . .
4| p . P . P . . .
3| . P . . . N . P
2| P . B N . . P .
1| . R . Q . R . K
   _ _ _ _ _ _ _ _
   A B C D E F G H

- It is Black's turn to move.
- No castling rights available.
- No en passant target square.
- Halfmove clock: 0
- Fullmove number: 18

I'm thinking about playing g7g6. Do you think I should play this move?

Assistant (SFT):
Moving the g-pawn to g6 looks harmless but immediately lets White eliminate the isolated
a-pawn with b3xa4. Black recovers only by shuffling the queen to a7 and the c8-bishop to d7,
gaining modest development while staying a pawn down. The pawn push also weakens the
dark-square complex around the king (g6/f5) and does nothing to address White's central
pressure on d5/e4. Overall the line leaves Black with less material and no clear
compensation – a clear downside.

Verdict: g7g6 is a suboptimal choice; avoid it in favor of a move that keeps the a-pawn or
improves piece activity without conceding material.
```

Figure 11: Sample of the *Guided Synthetic* data. Note that while the teacher model is provided with a line and centipawn difference, the teacher model is still prone to hallucination (it cites a material imbalance – this is not true). Additionally, despite prompting the teacher to use UCI notation, many examples still use SAN due to teacher model bias.

The **Verbalized Alpha-Beta Pruning** dataset is an entirely programmatically generated dataset built on top of Stockfish. This utilizes a program that does the following:

- For a given board, apply softmax with temperature sampling to choose initial move candidates (based on Stockfish move valuations).
- For each candidate, recursively build a tree that employs a similar softmax-based sampling algorithm.
- The recursion ends when a depth limit is reached, a max number of nodes are explored for this move, or a move is "written off" (below a quality threshold compared to other lines).
- Upon creation of the full move tree, each tree is verbalized using a separate algorithm that samples phrases from a large prompt bank to retain language diversity.
- The final winner (chosen via a minimax-based decision) is verbalized and used as the final answer.

We chose to include board valuations as well as minimax scores at decision nodes to instill a sort of value function $(V(s_t))$ in the model. Note that this value function is the absolute centipawn score – not a delta; we chose this to allow for verbalized minimax decision making. Further we instill several tree search strategies into the structure of the data – for example branching search, pruning, and minimax decision making.

The **Factual Board Answering** dataset generates multiple simple question-answer pairs about board states and combines them to ask multiple questions about the same board. Questions can ask about the legality of a proposed move, the location of pieces, whether or not a player is in check, the number of total legal moves available, and attacking piece information.

This dataset has the intention of training a model to explicitly learn piece and board dynamics in its latent space, with the hope that this latent ability will translate to downstream reasoning performance.

```
Verbalized Alpha-Beta Pruning Sample:

Assistant (SFT):
Interesting, let's consider some moves here.
<think> Ok, what if we played white queen moves to e1 (d1e1) [v+605]?
| From here they'll likely consider black bishop moves to g7 (f8g7) [v-66].
| | Ok, then we could play white pawn moves to a4 (a3a4) [v-56].
| They could also consider black pawn moves to h5 (h6h5) [v-71].
| | We could consider following with white knight moves to d5 (c3d5) [v-38].
| If the opponent played optimally, of the moves f8g7 and h6h5 they would likely choose
black bishop moves to g7 (f8g7) [mm-56] as this gives them the best position.
This position seems slightly better for the opponent [mm-56].

Let's think through white knight captures the pawn on e5 (f3e5) [v+787]. No, I don't like
this direction [mm-464].

How about we analyze the line starting with white rook moves to e1 (f1e1) [v+718]?
| A smart move from them could be black pawn moves to f5 (f6f5) [v+3].
| | We could consider following with white knight moves to d5 (c3d5) [v+4]. Instead we could
move white pawn captures the pawn on f5 (e4f5) [v+4]. Of all of these, we should play white
knight moves to d5 (c3d5) [mm+4].
The position after this appears essentially balanced [mm+4].

Our best move given how these would play out against optimal opponent play would be white
rook moves to e1 (f1e1). [mm+4] </think>

<answer> f1e1 </answer>
```

Figure 12: Sample of *Verbalized Alpha-Beta Pruning*. This sample highlights branching, minimax decision making, and an instance of pruning.

```
Factual Board Answering Sample:

User:
Here is a board in a game you're currently playing:
8| r . . . k . . r
7| p p . q b . p .
6| . . . p . . p p
5| . . . Q p . . .
4| . . P . P . b .
3| . . . . B . . .
2| P P . . . P . N
1| R . . . R . K .
  _ _ _ _ _ _ _ _
  A B C D E F G H

- It is White's turn to move.
- Castling rights: Black can castle kingside, Black can castle queenside.
- No en passant target square.
- Halfmove clock: 0
- Fullmove number: 20

Answer the following - if multiple questions, include a space between each answer:
Can your bishop take their pawn {'Yes', 'No'}?
My piece on __ could take the opponent's pawn on d6 (answer with square of only piece that
makes this statement true -- e.g., 'e4').
Is the black king in check {'Yes', 'No'}?
How many legal moves does your queen at d5 have (answer with an integer)?
Can you legally play d5g2 {'Yes', 'No'}?
Can your knight take their bishop {'Yes', 'No'}?

Assistant (SFT):
Yes d5 No 14 No Yes
```

Figure 13: Sample of the *Factual Board Answering* dataset.

The **Best Move** dataset asks the model to, given a board state, predict the best move directly with no chain of thought. This can be interpreted as learning a policy function ($\pi_\theta(a_t|s_t)$) via behavior cloning where the best move is suggested by a chess engine. One drawback of training on this data is that it can lead to very slow fine-tuning given each sample has $4$ trainable tokens.

Our final dataset – **Best Line** – is similar to *Best Move* except this asks to provide the line of optimal play (between 4 and 6 plies, chosen randomly to avoid a rigid structure), ending with a final estimate of the centipawn delta from this line. This is similar to *Guided Synthetic* and can be interpreted as verbalized $n$-step bootstrapping with $n = 2$ or $3$ dependent on the number of total plies.

Our intent is that this will instill a form of a world model through learning both transition dynamics and a value function. It can also be viewed as an extension of *Best Move* that condenses multiple optimal move data-points into a single sample – this is beneficial from a training efficiency perspective as well.

```
Best Line Sample:

User:
{Visual (ASCII) board format}

What are the likely next 6 plies that would play out? List all moves in UCI notation (e.g.,
e4f6) separated by spaces. If a checkmate occurs, end with 'mate'. Otherwise, end with the
change in centipawns in the format '[Δ+/-#]'.

Assistant (SFT):
e8c8 f3d2 h5g6 d1e1 g6c2 d2c4 [Δ-21]
```

```
Best Move Sample:

User:
{Visual (ASCII) board format}

Respond immediately with just the move in UCI notation (e.g. 'e2e4') -- nothing else.

My question:
What is the best move?

Assistant (SFT):
f4g2
```

Figure 14: Samples of *Best Line* and *Best Move*.

## D    DATA INCLUSION ANALYSES

First, we outline the various data inclusion analyses we ran. The purpose was to understand which datasets were most effective to inform our final scaled experiments – see Appendix C for detail and examples for each dataset. Figure 15 outlines our experiments, including the token distributions for each dataset used in SFT. Our results (SFT and RL) on each data mix are included in Tables 1 and 2. Additionally, Table 3 includes results on tasks that are not trained on during the RL stage.

| # in mm | Rejection Sampling [RSPM] (Only Predict Move) | Rejection Sampling [RSA] (All Evals) | Verbalized Alpha-Beta Pruning [VABP] | Factual Board Answering [FBA] | Guided Synthetic [GS] | Best Move [BM] | Best Line [BL] | Magpie [MGP] | Total Tokens |
|---|---|---|---|---|---|---|---|---|---|
| **Inclusion Experiments** | | | | | | | | | |
| SFT1 [RSPM] | 2.50 | - | - | - | - | - | - | 5.00 | **7.50** |
| SFT2 [RSA] | - | 5.00 | - | - | - | - | - | 2.50 | **7.50** |
| SFT3 [VABP] | - | 3.75 | 1.88 | - | - | - | - | 1.88 | **7.50** |
| SFT4 [FBA] | - | 3.75 | - | 1.88 | - | - | - | 1.88 | **7.50** |
| SFT5 [GS] | - | 3.75 | - | - | 1.88 | - | - | 1.88 | **7.50** |
| SFT6 [BM] | - | 3.75 | - | - | - | 1.88 | - | 1.88 | **7.50** |
| SFT7 [BL] | - | 3.75 | - | - | - | - | 1.88 | 1.88 | **7.50** |
| SFT8 [BM - All] | - | 2.25 | 0.75 | 0.75 | 0.75 | 2.25 | - | 0.75 | **7.50** |
| SFT9 [BL - All] | - | 2.25 | 0.75 | 0.75 | 0.75 | - | 2.25 | 0.75 | **7.50** |
| **Scaled Runs** | | | | | | | | | |
| SFT8 XL | - | 7.50 | 3.80 | 2.20 | 7.50 | 30.00 | - | 9.00 | **60.00** |
| SFT9 XL | - | 7.00 | 3.50 | 4.00 | 7.50 | - | 30.00 | 8.00 | **60.00** |
| SFT8 + SFT9 XL | - | 14.50 | 7.30 | 6.20 | 15.00 | 30.00 | 30.00 | 17.00 | **120.00** |

Figure 15: Distribution of tokens used in each experiment. Token numbers are shown in millions; we sampled our data to match this distribution, though there may be immaterial variations for actual token counts used. We include tags (e.g., [VABP]) for mnemonic reference. Note that with the *Rejection Sampling (All Evals)* [RSA] dataset, we allocate $50\%$ of tokens to the *Predict Move* task and sample the remainder from the other evaluation tasks. `SFT8 + SFT9 XL` was trained by taking the `SFT8 XL` model checkpoint and training on the `SFT9 XL` dataset.

Table 1: Results are shown for the `Predict Move` evaluation on 400 samples. *Predict Move Average Rank* is the average normalized rank (with 0 being the worst move and 1 being the best move per Stockfish) of the legal moves provided in this task.

| Experiment Name | SFT Train Tokens | RL[a] Samples | Pred. Move % Legal ↑ | | Pred. Move Avg. Rank ↑ | | % Trivial[b] Moves ↓ | |
|---|---|---|---|---|---|---|---|---|
| | | | SFT | RL | SFT | RL | SFT | RL |
| **Baselines** | | | | | | | | |
| Qwen2.5 7B-Instruct | – | – | 8% | | 0.56 | | 6% | |
| Llama 4 Maverick | – | – | 42% | | 0.62 | | **1%** | |
| gpt-oss-120b (Medium) | – | – | **94%** | | **0.66** | | 2% | |
| **Inclusion Experiments** | | | | | | | | |
| SFT1 [RSPM] | 15M | 8k[a] | 34% | 82% | 0.63 | 0.52 | 4% | 55% |
| SFT2 [RSA] | 15M | 8k | 37% | 73% | 0.63 | 0.60 | 5% | 48% |
| SFT3 [VABP] | 15M | 8k | 40% | 67% | 0.61 | 0.53 | 3% | 46% |
| SFT4 [FBA] | 15M | 8k | 44% | 72% | 0.59 | 0.54 | 3% | 40% |
| SFT5 [GS] | 15M | 8k | 36% | 71% | 0.60 | 0.54 | 3% | 42% |
| SFT6 [BM] | 15M | 8k | 44% | **93%** | **0.64** | **0.71** | 4% | 31% |
| SFT7 [BL] | 15M | 8k | 48% | 85% | 0.62 | 0.63 | **2%** | 64% |
| SFT8 [BM - All] | 15M | 8k | **60%** | 90% | 0.60 | **0.71** | 3% | **14%** |
| SFT9 [BL - All] | 15M | 8k | 51% | 89% | 0.60 | 0.64 | 7% | 32% |
| **Scaled Runs** | | | | | | | | |
| SFT8 XL | 60M | 16k | **55%** | **98%** | **0.62** | 0.82 | 3% | **12%** |
| SFT9 XL | 60M | 16k | 52% | 93% | 0.60 | 0.75 | 3% | 25% |
| SFT8 + SFT9 XL[c] | 120M | 16k | 54% | **98%** | 0.58 | **0.83** | **2%** | 22% |

[a] All experiments used equal portions of the four evaluation types for RL except SFT1 which trained on 8k samples of only `Predict Move`.

[b] "Trivial Moves" consist of edge pawn moves (e.g., `a2a4`, `a2a3`) or king/rook wiggles (e.g., `a1b1`, `b1a1`). These were chosen based on identified reward hacking behaviors.

[c] For this run we trained the `SFT8 XL` checkpoint with the `SFT9 XL` dataset.

Table 2: See Figure 15 for detail on the data included in each inclusion experiment and Table 1 for performance on the `Predict Move` task. Each evaluation shown is based on 400 unique samples for each task. The *Legal Moves* task asks the model to produce a list of legal moves given a target piece – the score is measured as intersection over union (IoU) vs. ground truth. *Best Move* and *Worst Move* ask the model to, given a list of 5 moves, choose the best (or worst, respectively) move of the list – the incorrect candidates are sampled such that they are beyond a sufficient threshold of difference per Stockfish. See Appendix B for examples of each task.

| Experiment Name | Legal Moves IoU ↑ | | Best Move Acc. ↑ | | Worst Move Acc. ↑ | | Ref'd Move Acc.[a] ↑ | | Avg. Reas. Quality[b] ↑ | |
|---|---|---|---|---|---|---|---|---|---|---|
| | SFT | RL | SFT | RL | SFT | RL | SFT | RL | SFT | RL |
| **Baselines** | | | | | | | | | | |
| Qwen2.5 7B-Instruct | 0.26 | | 19% | | 21% | | 12% | | 6.4 | |
| Llama 4 Maverick | 0.43 | | 27% | | 31% | | 38% | | 6.6 | |
| gpt-oss-120b (Medium) | **0.96** | | **57%** | | **79%** | | **70%** | | **7.0** | |
| **Inclusion Experiments** | | | | | | | | | | |
| SFT1 [RSPM] | 0.26 | 0.17 | 23% | 19% | 25% | 23% | 33% | 76% | 5.4 | 4.7 |
| SFT2 [RSA] | 0.37 | 0.44 | 29% | 34% | 30% | 48% | 35% | 57% | 5.3 | 4.2 |
| SFT3 [VABP] | 0.41 | 0.58 | 25% | 35% | 30% | 48% | 34% | 45% | 5.3 | 2.3 |
| SFT4 [FBA] | **0.49** | 0.58 | 26% | 36% | 30% | 51% | 39% | 63% | 5.0 | 4.0 |
| SFT5 [GS] | 0.37 | 0.57 | 30% | 35% | 29% | 54% | 35% | 52% | 5.3 | **6.1** |
| SFT6 [BM] | 0.42 | 0.66 | 29% | 37% | 32% | 45% | 41% | **86%** | 5.3 | 2.2 |
| SFT7 [BL] | 0.46 | 0.63 | 25% | 28% | 32% | 51% | 38% | 68% | 5.0 | 2.4 |
| SFT8 [BM - All] | 0.44 | **0.67** | 31% | **41%** | 34% | 55% | 53% | 80% | **5.7** | 2.9 |
| SFT9 [BL - All] | 0.40 | 0.59 | 29% | 38% | 28% | 53% | 44% | 80% | 5.6 | 3.5 |
| **Scaled Runs** | | | | | | | | | | |
| SFT8 XL | 0.46 | 0.79 | 29% | 60% | **33%** | 58% | **48%** | 83% | 5.7 | 2.2 |
| SFT9 XL | 0.47 | 0.75 | 28% | 57% | **33%** | 62% | **48%** | 88% | 5.7 | **5.3** |
| SFT8 + SFT9 XL | **0.52** | **0.87** | 31% | **62%** | 31% | **64%** | 47% | **90%** | 5.8 | 5.3 |

[a] *Referenced Move Accuracy* is measured by using Llama 4 Maverick to parse reasoning outputs and create a list of all moves that are mentioned by the model during the reasoning trace. These moves are then run through a chess engine to determine what percent are legal as a measure of reasoning factuality. Appendix E has further detail on measuring hallucinations.

[b] *Average Reasoning Quality* is measured by using gpt-oss-120b as a judge and is the simple average of scores provided for *Reasoning Efficacy*, *Reasoning Efficiency*, and *Reasoning Faithfulness*. Further detail is provided in Appendix G on measuring reasoning quality.

***Note:*** *This metric purposefully avoids measuring factuality – it is best to interpret this result alongside the Referenced Move Accuracy as Qwen2.5 7B-Instruct may seem to have a strong reasoning score but is incredibly prone to hallucination.*

Table 3: See Figure 15 for detail on the data included in each inclusion experiment. Results are testing performance on tasks not trained on during the RL stage. Note that some of the experiments were trained on FBA data during the SFT stage – these are tagged with footnote c.

| Experiment Name | Factual Board Answering[a]↑ | | Out-of-Distrib. Mates[b]↑ | |
|---|---|---|---|---|
| | SFT | RL | SFT | RL |
| **Baselines** | | | | |
| Qwen2.5 7B-Instruct | 31.6% | | 0.0% | |
| Llama 4 Maverick | 46.7% | | 14.0% | |
| gpt-oss-120b (Medium) | **99.5%** | | **78.3%** | |
| **Inclusion Experiments** | | | | |
| SFT1 [RSPM] | 38.2% | 36.8% | 4.2% | 0.7% |
| SFT2 [RSA] | 40.6% | 32.2% | 3.5% | 2.0% |
| SFT3 [VABP] | 41.1% | 34.3% | 4.0% | 0.8% |
| SFT4 [FBA][c] | **59.8%** | **58.0%** | 6.3% | 3.7% |
| SFT5 [GS] | 41.7% | 35.7% | 3.7% | 2.0% |
| SFT6 [BM] | 36.5% | 34.0% | 6.5% | 6.0% |
| SFT7 [BL] | 40.2% | 37.2% | 5.8% | 8.2% |
| SFT8 [BM - All][c, d] | 36.8% | 36.8% | 6.2% | 5.0% |
| SFT9 [BL - All][c] | 57.1% | 57.9% | **9.3%** | **13.5%** |
| **Scaled Runs** | | | | |
| SFT8 XL[c] | 39.5% | 71.1% | 7.0% | 5.3% |
| SFT9 XL[c] | **60.9%** | 64.7% | 8.3% | 8.5% |
| SFT8 + SFT9 XL[c] | 59.3% | **82.3%** | **10.7%** | **14.8%** |

[a] *Factual Board Answering (FBA)* average score is on 1,000 unseen FBA tasks sampled equally across five problem types. All task scores $\in [0, 1]$ with 1 being the highest score possible.

[b] *Out-of-Distribution Mates* average score is on 600 tasks sampled evenly across three problem types. Problems are sourced from Mészáros et al. (2025), specifically the *Knights & Rooks*, *More Pieces*, and *Same Color* problem types. See Figure 16 for example problems of each task.

[c] Experiments included *Factual Board Answering* data in the SFT stage. All evaluations are on unseen tasks.

[d] Both the SFT and RL checkpoints received 0.0% on two of the FBA tasks due to failure to follow instructions.

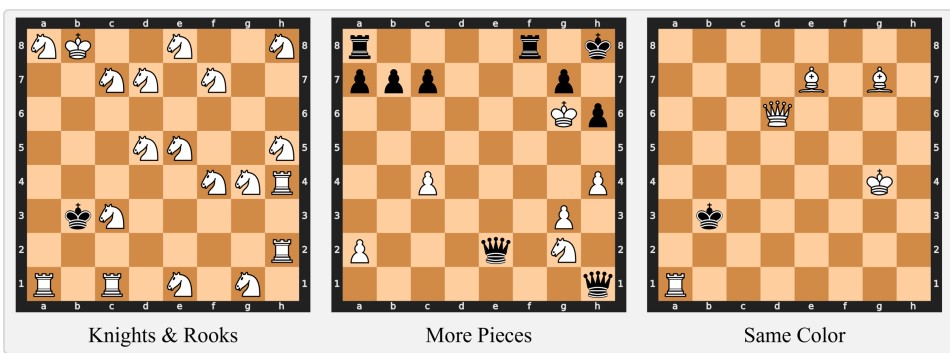

Knights & Rooks       More Pieces       Same Color

Figure 16: Examples from each of the three subtasks included in the *Out-of-Distribution Mates* evaluation set sourced from Mészáros et al. (2025). The task is to play a checkmate given the position – there may be multiple valid checkmates, providing any results in a correct answer.

# E HALLUCINATIONS

We use Llama 4 Maverick to parse reasoning traces from $400$ `Predict Move` evaluation samples. For each sample, the parsing generates two lists:

- **Moves**: This is a list of all moves referenced by the model in its reasoning trace played by the player.
- **Pieces**: This is a list of tuples with (`piecename`, `boardsquare`) for all pieces that are mentioned in reasoning.

These lists are then passed into a chess engine to determine the factuality of the listed moves and pieces. *Mean Total Reasoning Accuracy* is computed as the sum of correct moves and correct pieces divided by the total number of provided moves and pieces – hallucination rate can simply be computed with $(1 - Accuracy)$.

*Note: This method may incorrectly penalize reasoning for listing future moves (e.g., play a2a4 followed by a4a5) or legal moves that the opponent may play. However, in review we found these to be rare in occurrence.*

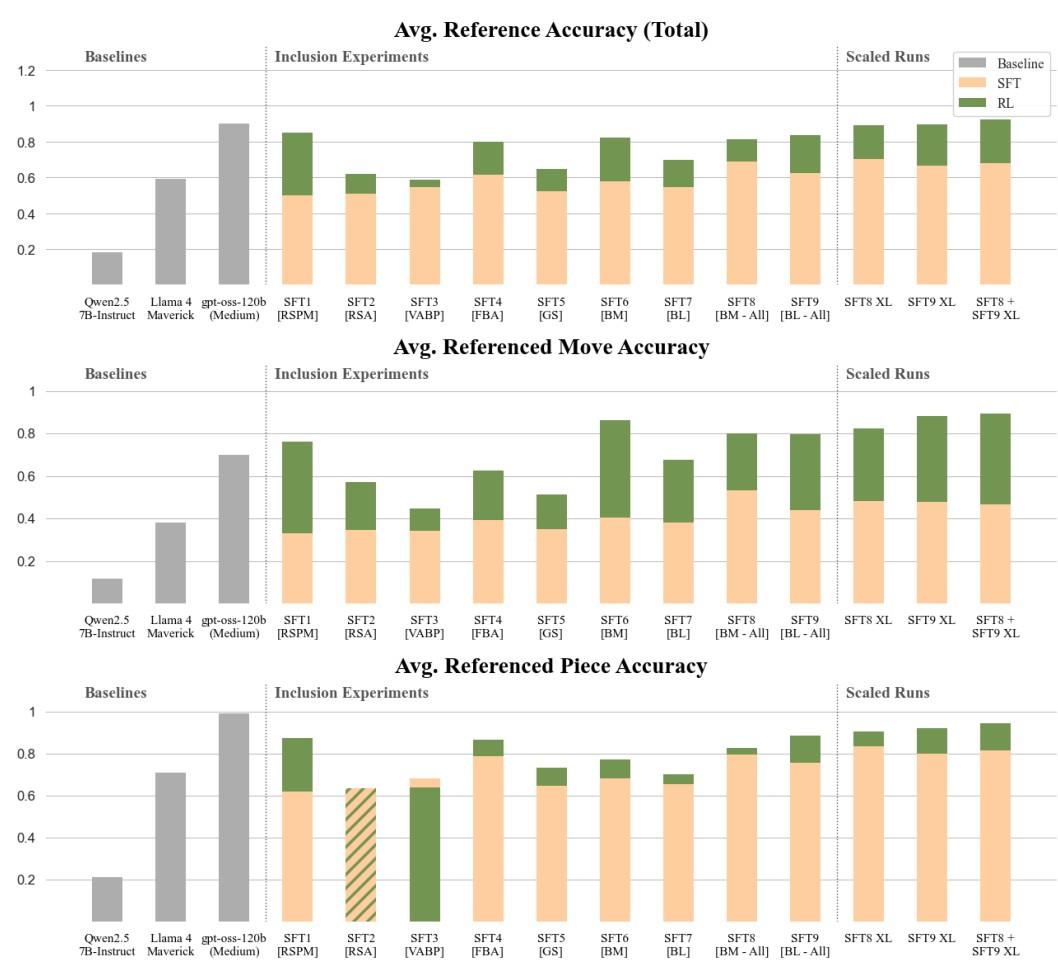

Figure 17: Accuracy of tested models for both moves and pieces referenced in their reasoning traces. Bars are overlaid directly on top of each other and stacking is not cumulative. Accuracy is computed as the number of correct references divided by the total number of references. See Figure 15 for detail on the data included in each experiment. Hatched lines are shown in cases where the SFT and RL runs are within $2\%$ of each other.

# F REASONING STRATEGIES

Figure 18 highlights the usage of various reasoning strategies across tested models. We follow from Gandhi et al. (2025) and Zeng et al. (2025), and we also include two other strategies in *Self-Correction* (the model explicitly corrects something stated previously) and *Tree Search*. See Figure 15 for detail on the data included in each experiment.

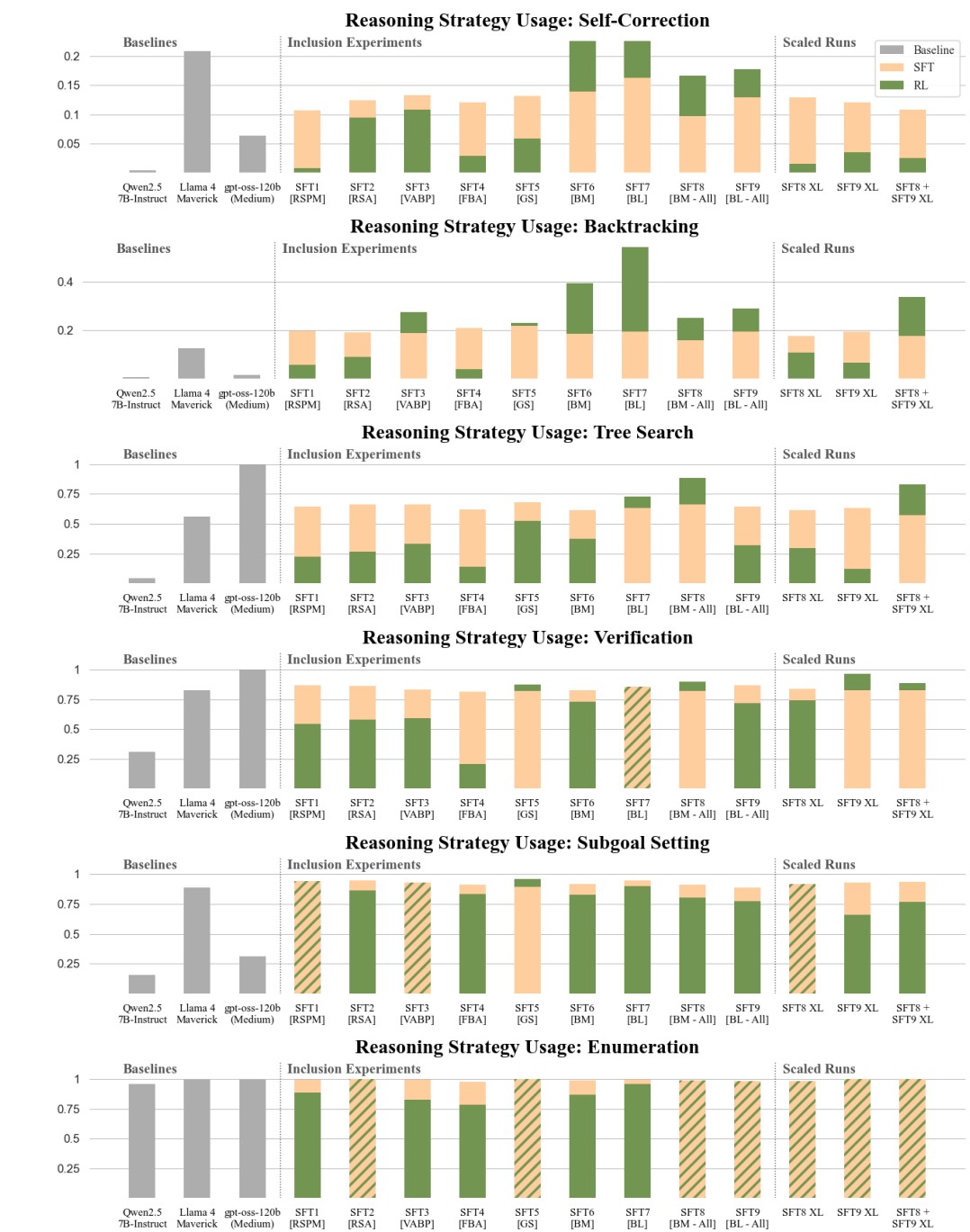

Figure 18: Usage rate of reasoning strategies on $400$ `Predict Move` tasks. Bars are overlaid directly on top of each other and stacking is not cumulative. Reasoning strategies are parsed using Llama 4 Scout and usage is measured as a binary flag for each evaluation sample. Hatched lines are shown in cases where the SFT and RL runs are within $2\%$ of each other.

# G  REASONING QUALITY

To analyze reasoning quality, we employ LLM-as-a-judge Zheng et al. (2023) using gpt-oss-120b. Table 4 highlights the results from a statistical correlation analysis between LLM-judge outputs and expert scores (authors) – all metrics are statistically significant. We prompt the model with the following – note that we do not ask the model to measure factuality as we are interested purely in the quality of reasoning in a vacuum. Please refer to Appendix E for detail on hallucination rates and see Figure 15 for detail on the data included in each experiment.

> **Reasoning Quality Judge Instructions:**
>
> *Your task is to be a critical judge.*
>
> *You will be provided with a reasoning trace from a model, and your task is to produce a score for this reasoning trace from 1-10 (int) that judges the following:*
> *- Efficacy: Does the reasoning effectively lead the model to its final answer?*
> *- Efficiency: Is the reasoning efficient and targeted?*
> *- Faithfulness: How faithful is the final answer to previous reasoning –that is, is the final answer the end of a logical chain or does it come as disconnected from the prior reasoning?*
> *...*

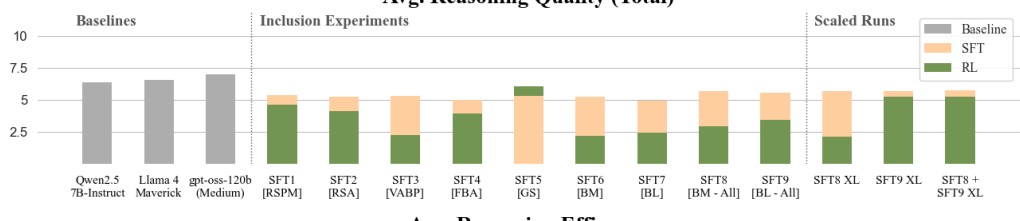

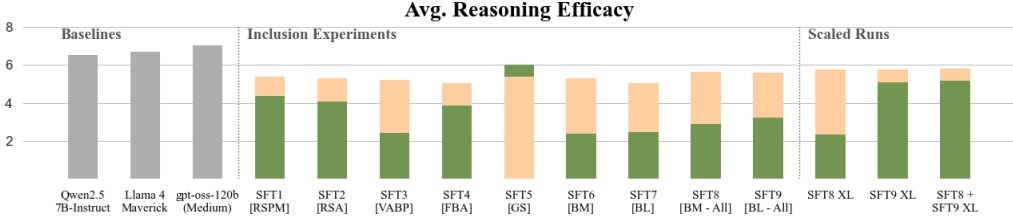

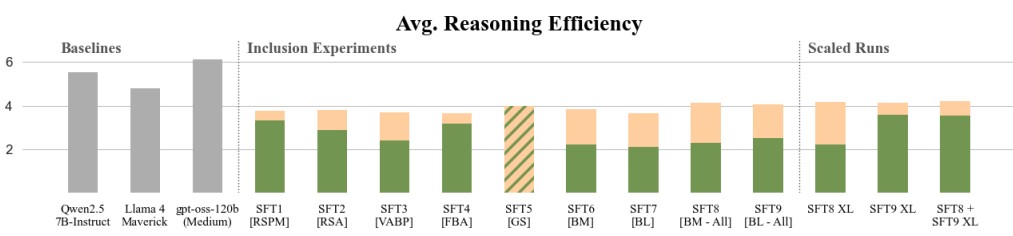

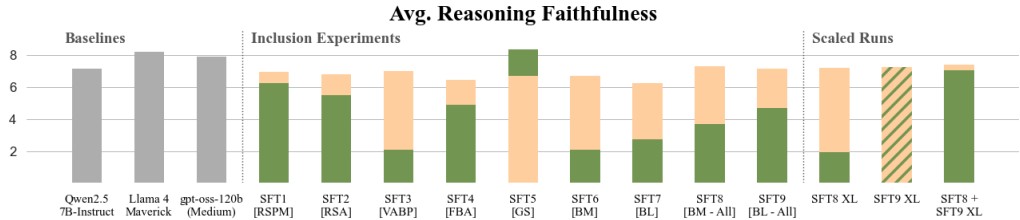

Figure 19: Reasoning quality scores on 400 `Predict Move` tasks. Bars are overlaid directly on top of each other and stacking is not cumulative. Reasoning quality is scored by gpt-oss-120b and scores are provided from 1 to 10. The *Mean Reasoning Quality (Total)* score is a simple average over the three subcategories. Hatched lines are shown in cases where the SFT and RL runs are within 2% of each other.

Additionally, we provide an example of *unfaithful* reasoning that earns a 1 out of 10 score on reasoning faithfulness. This sample is from our scaled *Best Move - All* final RL model.

Table 4: Statistical significance of expert and LLM-judge scores across qualitative reasoning metrics. Samples are graded blindly and drawn randomly from among all tested models (baselines and experiments) excluding gpt-oss-120b. We exclude gpt-oss-120b as language models have shown a tendency to favor their own outputs (Panickssery et al., 2024) which may adversely affect our analysis.

| Expert | N | Efficacy | | Efficiency | | Faithfulness | | Sum of All | |
|---|---|---|---|---|---|---|---|---|---|
| | | Corr | P-Val | Corr | P-Val | Corr | P-Val | Corr | P-Val |
| Expert 1 | 96 | 0.41 | 4.01e-05 | 0.31 | 2.03e-03 | 0.68 | 1.50e-14 | 0.57 | 1.10e-09 |
| Expert 2 | 73 | 0.24 | 4.00e-02 | 0.34 | 3.07e-03 | 0.43 | 1.71e-04 | 0.39 | 6.63e-04 |

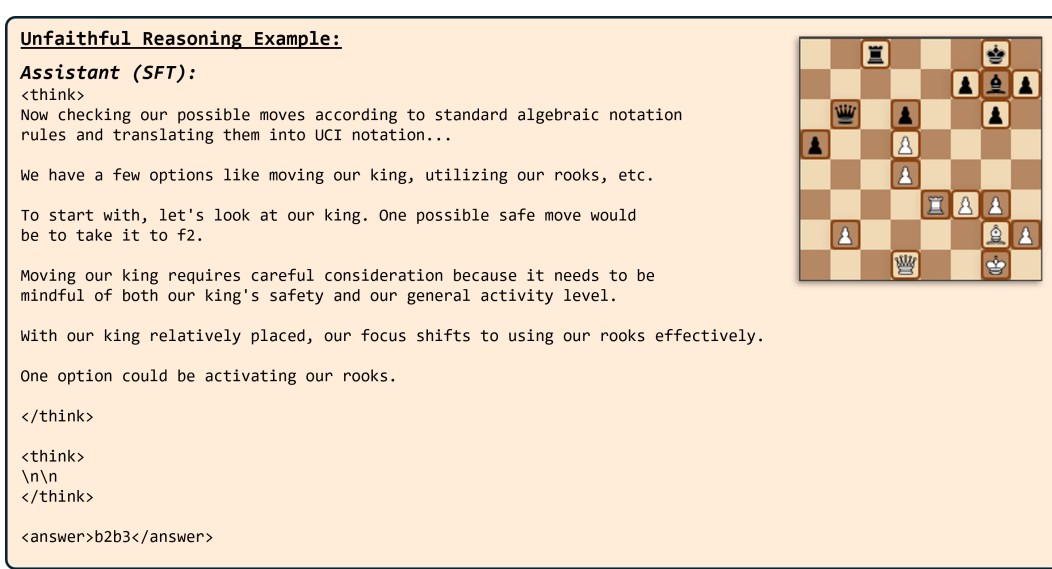

**Unfaithful Reasoning Example:**

***Assistant (SFT):***
```
<think>
Now checking our possible moves according to standard algebraic notation
rules and translating them into UCI notation...

We have a few options like moving our king, utilizing our rooks, etc.

To start with, let's look at our king. One possible safe move would
be to take it to f2.

Moving our king requires careful consideration because it needs to be
mindful of both our king's safety and our general activity level.

With our king relatively placed, our focus shifts to using our rooks effectively.

One option could be activating our rooks.

</think>

<think>
\n\n
</think>

<answer>b2b3</answer>
```

Figure 20: Example of unfaithful reasoning – given a score of 1 in reasoning faithfulness. Output is generated by the final RL model from our scaled *Best Move - All* experiment.

# H   SFT AND RL HYPERPARAMETERS

See Tables 5 and 6 for training hyperparameters.

All experiments were run on Nvidia A100 or H100 chips. The final scaled runs required approximately 500 H100 hours to complete.

Table 5: SFT training hyperparameters.

| Parameter | Value |
| --- | --- |
| Training engine | LlamaFactory (Zheng et al., 2024) |
| Fine-tuning type | Full SFT |
| LR scheduler | Cosine |
| Precision | BF16 |
| Optimizer | AdamW (Loshchilov & Hutter, 2019) |
| Learning rate | $3 \times 10^{-6}$ |
| Warmup ratio | 0.1 |
| Train batch size | 64 |
| Training data (tokens $\times$ epochs) | $7.5mm \times 2$ (inclusion tests) |
|  | $60mm \times 1$ (scaled runs) |

Table 6: RL training hyperparameters.

| Parameter | Value |
| --- | --- |
| Training engine | veRL (Sheng et al., 2025) |
| Objective | Dr. GRPO (Liu et al., 2025) |
| Learning rate | $1 \times 10^{-6}$ |
| Train batch size | 64 |
| Max response length | $3,000$ tokens |
| Actor clip ratio (low/high) | 0.20 / 0.28 (Yu et al., 2025) |
| Use KL loss | False (off) |
| Rollouts per sample | 8 |
| Entropy coefficient | 0 (off) |
| Number of samples | $8,192$ unique samples (inclusion tests) |
|  | $16,384$ unique samples (scaled runs) |

# I REINFORCEMENT LEARNING TRAINING DYNAMICS

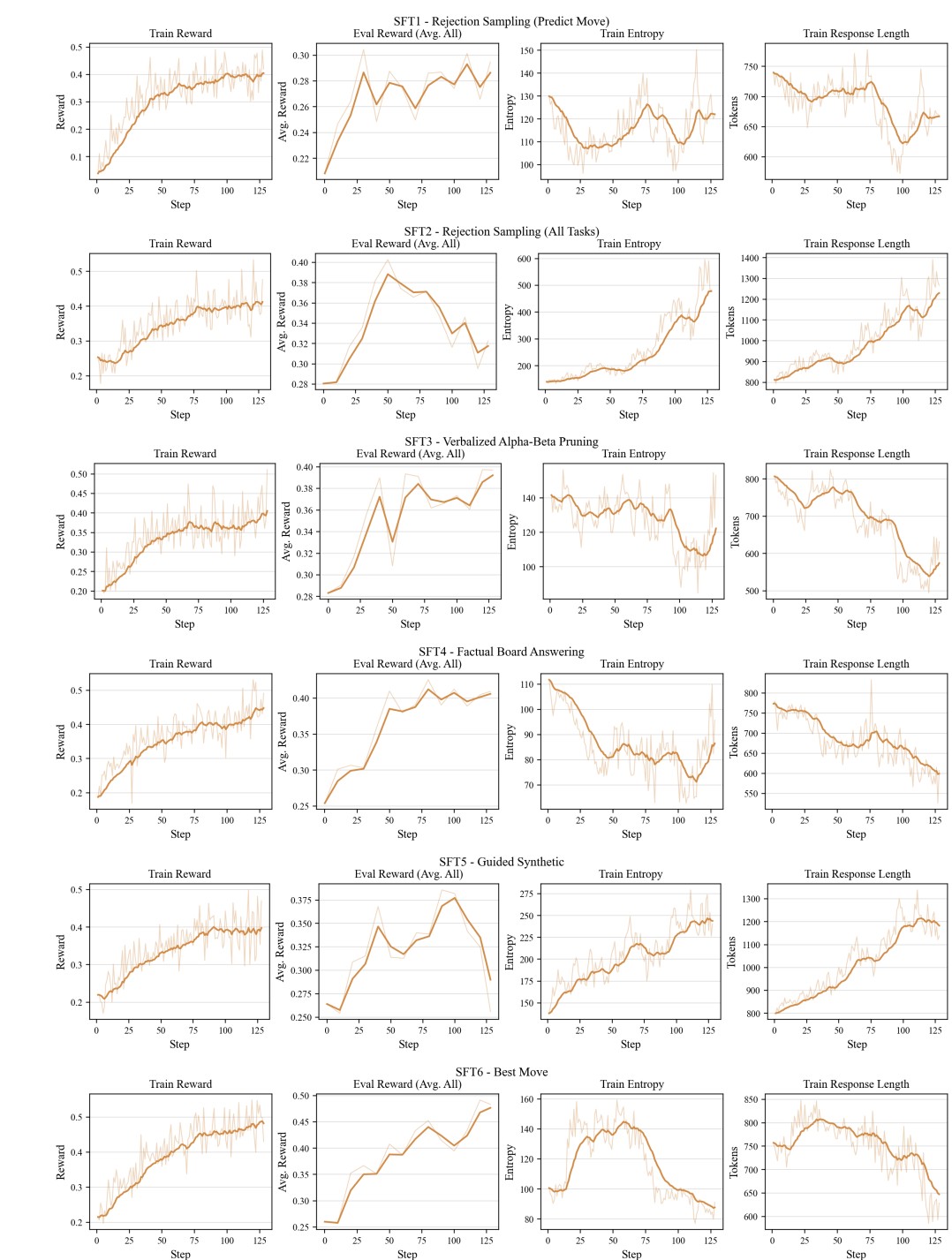

Figure 21: Select training dynamics during reinforcement learning across our inclusion and scaled experiments.

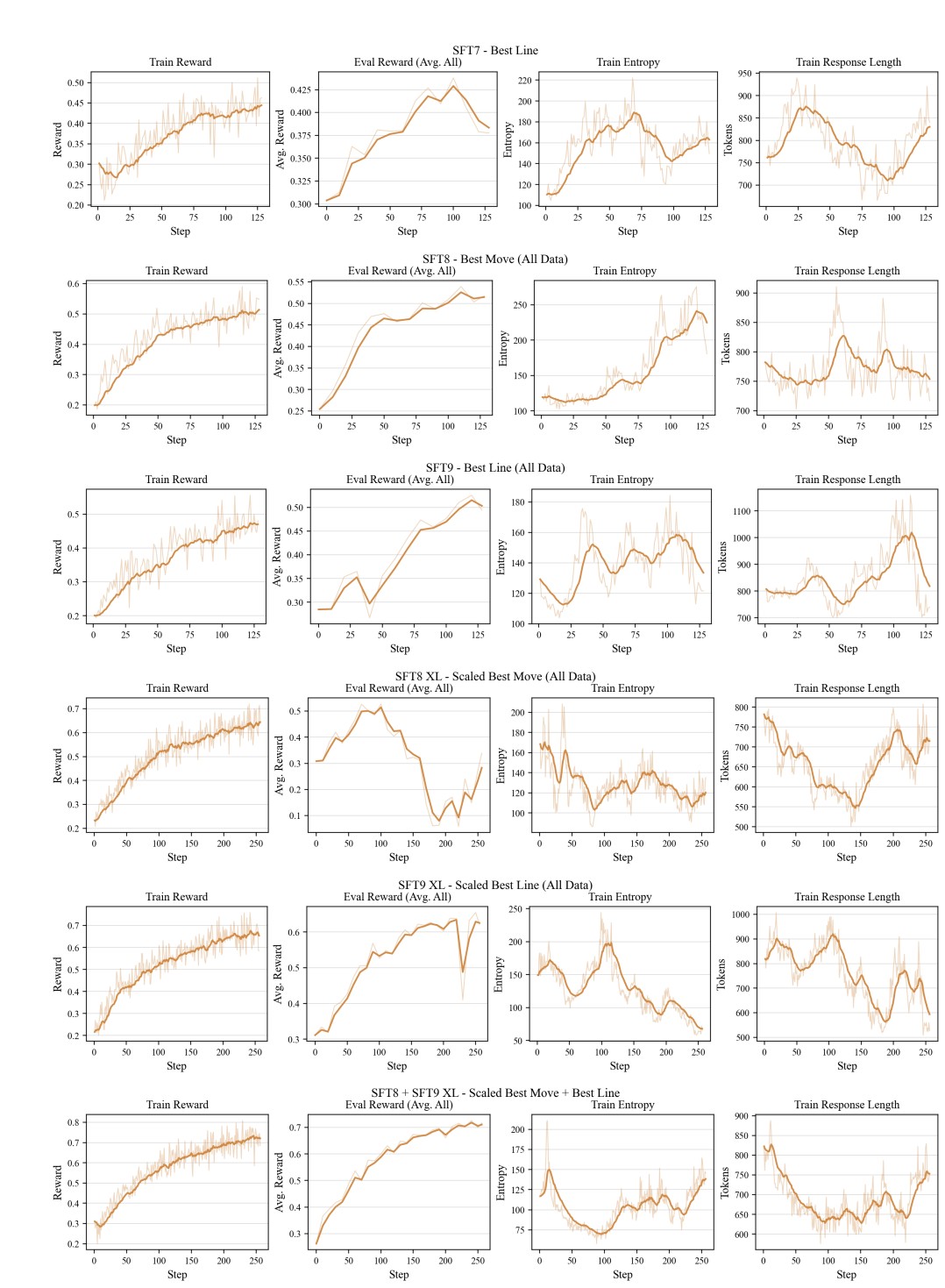

Figure 21: Select training dynamics during reinforcement learning across our inclusion and scaled experiments (cont.).

