# OpenReview forum: "How Reasoning Evolves from Post-Training Data in Sequential Decision-Making Domains"
_ICLR.cc/2026/Conference — Submitted to ICLR 2026_

### Official Review · Reviewer_sfa6 · 2025-10-29

**Soundness:** 4
**Presentation:** 4
**Contribution:** 3
**Rating:** 6
**Confidence:** 3

**Summary:**

This paper investigates how reasoning emerges and develops in large language models fine-tuned for sequential decision-making tasks, specifically using chess as a verifiable testbed. The authors construct and analyze a suite of datasets grounded in theoretical motifs (Best Move, Best Line, Verbalized Alpha-Beta, etc.), study the effects of dataset composition on SFT, and follow with RL to assess how model behaviors, reasoning faithfulness, move quality, and hallucination rates evolve.

**Strengths:**

**1. Originality:** The systematic comparison of six dataset construction methods (Best Move, Best Line, Factual Board Answering, Verbalized Alpha-Beta Pruning, Rejection Sampling, Guided Synthetic) across both SFT and RL phases is novel. Using chess as a verifiable reasoning domain is well-motivated, enabling precise measurement of reasoning quality.

**2. Quality:** Experimental design is rigorous and comprehensive. Evaluation is multifaceted, combining quantitative metrics (move accuracy, legal move rate, average move rank) with qualitative analysis (reasoning strategy usage, hallucination rates, faithfulness).

**3. Clarity:** Great presentation throughout. Claims are well-supported by clear visualizations. The systematic organization makes the comparative analysis easy to follow and findings straightforward to interpret.

**Weaknesses:**

**1. Limited Generalizability:** Experiments use only Qwen2.5-7B-Instruct on chess. Key findings—especially the faithful/unfaithful reasoning distinction—require validation on alternative base models and domains (math, coding) to establish whether they represent fundamental reasoning principles or chess/model-specific artifacts.

**2. Insufficient Training Dynamics Analysis:** The work primarily reports task accuracy (training rewards), overlooking critical training signals (response length, entropy, or KL divergence) that would provide mechanistic insights into observed phenomena like RL stability differences and unfaithful reasoning emergence. Deeper analysis of training dynamics would substantially strengthen claims and broaden applicability.

**Questions:**

1. Can the authors provide more detail or statistical analysis regarding the faithfulness/quality metrics? For example, can the reliability of the LLM-as-a-judge scores be compared to human expert assessment for a sample of traces?

2. The authors show Best Move achieves comparable performance to Best Line despite unfaithful reasoning. This raises concerns: Is faithful reasoning actually necessary for strong chess performance? What are the implications for reasoning models more broadly if unfaithful reasoning suffices?

3. While hallucination rates drop post-RL and unfaithful rationalization is noted, more granularity is needed: What are the most persistent failure modes? Do failure modes remain unchanged during SFT and RL?

---

> ### Author Response · Authors · 2025-11-21
> **Response**
>
> Thanks for reviewing! Below are some responses:
>
> >Weakness #1: Limited Generalizability: Experiments use only Qwen2.5-7B-Instruct on chess. Key findings—especially the faithful/unfaithful reasoning distinction—require validation on alternative base models and domains (math, coding) to establish whether they represent fundamental reasoning principles or chess/model-specific artifacts.
>
> We are currently finalizing a few experiments generalizing this to other domains. Specifically we're doing this in AlfWorld and CookingWorld which are text-based domains that are tied to embodied AI tasks -- the intent here is to show that in broader sequential decision making domains these results hold.
>
> >Weakness #2:  Insufficient Training Dynamics Analysis: The work primarily reports task accuracy (training rewards), overlooking critical training signals (response length, entropy, or KL divergence) that would provide mechanistic insights into observed phenomena like RL stability differences and unfaithful reasoning emergence. Deeper analysis of training dynamics would substantially strengthen claims and broaden applicability.
>
> Good idea here -- we are including a new appendix with this data for all our runs. It was an area that we largely didn't extract many insights from but as we have the data it would be beneficial to add.
>
>
> >Question #1: Can the authors provide more detail or statistical analysis regarding the faithfulness/quality metrics? For example, can the reliability of the LLM-as-a-judge scores be compared to human expert assessment for a sample of traces?
>
> Reviewer 1Npe had a similar question so we went ahead and conducted a statistical analysis vs. human experts. Note that this is something we conducted a lot of prompt tuning on and scrutinized heavily during our work. We agreed this is a valuable addition to understand so we ran an experiment using 2 human experts (see the general comment above) -- we see our qualitative measurements are statistically significant along all dimensions.
>
> Couple things we'd like to note about these results:
> - Some of the reasoning can be really noisy so grading is tricky -- especially for poor performers or overly verbose / uninterpretable traces (e.g., gpt-oss-120b). We do feel confident, especially with the statistical results, that there is useful signal in these qualitative measurements.
> - Grading faithfulness is much easier -- "How well does the final move align with what reasoning was discussing?" It still is noisy, but as we see there is much higher agreement with experts here and we feel this measurement is warranted as it underlies several of our larger takeaways.
>
> >Question #2: The authors show Best Move achieves comparable performance to Best Line despite unfaithful reasoning. This raises concerns: Is faithful reasoning actually necessary for strong chess performance? What are the implications for reasoning models more broadly if unfaithful reasoning suffices?
>
> Yep! This result is a super valid question and we agree it is quite unintuitive…this actually motivates some of our future work :)
>
> We believe this unfaithful reasoning could point to a couple things:
> 1) Unnecessary reasoning length (i.e., the model ‘knows’ its final answer early and the reasoning is unnecessary). If you look at the recent improvements in GPT 5.1 and Grok 4.1 you’ll notice that a key thing they pushed is using thinking efficiently — hard problems deserve more tokens and easy problems should be quickly answered — and across most benchmarks these models are improvements.
> 2) It also has many implications on LLM safety and interpretability as well; we cite Turpin, 2023 that was some of the earlier work uncovering that models may not always act as their reasoning implies (unfaithful reasoning).
> 3) This may also imply that verbalized reasoning (discrete token space) may not be the most effective test-time compute method — rather something like Coconut (continuous space reasoning) [Hao, 2024] that we also cite could be an effective direction to further study.
>
> All of which saying that these results opened up new research questions for us!

---

> > ### Author Response · Authors · 2025-11-21
> > **Response (cont.)**
> >
> > >Question #3: While hallucination rates drop post-RL and unfaithful rationalization is noted, more granularity is needed: What are the most persistent failure modes? Do failure modes remain unchanged during SFT and RL?
> >
> > Interesting point -- it would be great to have clear understanding of what the most common failure nodes are as directly addressing these would lead to clear improvements. We don't have this data quantified and it would be a much larger scope to do so, but we can provide some guidance based on all the reasoning traces we've read:
> > - Commonly, the weaker models just hallucinate a lot. Addressing hallucinations was a motivation for several of our synthetic datasets as even the rejection sampling data was very prone to hallucinations.
> > - You see repetitive themes in reasoning despite it not being as relevant for the current position of the board. Some of the models will always default to mentioning common chess-speak like 'I want to control the center of the board' even in situations when that is not necessarily the best approach.
> > - Some reward hacking. We saw this and alleviated some of it, but our models tend to be on the 'cautious' side of playing moves. You see this with some of the 'trivial moves played' results in Appendix D. We believe that moving to a multi-turn RL setting that we discuss in the limitations section would reduce this.
> >
> > These are a few examples that we have seen directly. Fixing this would require better SFT data (especially in lieu of the rejection sampling data we used from Llama 4 Maverick). Some of this would require adjusting the RL training steps (we mention this in the limitations section). We're not trying to hide any of this information and would be happy to elaborate a bit more in the limitations section if you believe that would be beneficial!

---

### Official Review · Reviewer_1Npe · 2025-10-30

**Soundness:** 2
**Presentation:** 2
**Contribution:** 2
**Rating:** 4
**Confidence:** 4

**Summary:**

The authors look to determine good decision decisions and takeaways on developing a reasoning model using chess as a testbed. They train Qwen 7B over a range of different dataset SFT and then RL.

**Strengths:**

* The finding that best line works well is interesting.("Best Line: Given a board, predict the optimal line of play (4 − 6 plies) ending with the expected centipawn delta from playing this line.") It offers some insight toward efficiently training LM-style models.
* I like the structure of the tables in the appendix, I think it is effective.

**Weaknesses:**

* Language models so far aren't great at playing chess directly and trained model does not exceed/match the performance of the 120B model so it is unclear how well these findings scale. Or how meaningful they are.
* This work does not reconnect the training results with downstream/tournament/chess results. Further grounding the results w.r.t. to external models or performance would help position this work.
* I think the reasoning faithfulness metrics were not well justified (see questions)
* It took a whole to understand how/what data mixtures were used. Part of it is because Figure 4 really doesn't capture the information and "Best Line" and "Best Move" mixtures include other data as well. I think moving some of the information up from APP D DATA INCLUSION ANALYSES would help set this straight. Also, I think it is reasonable to do, but the final best run Best Move + Best Line also has 2x the data of other runs. (Which makes it unclear how you did the comparison for Figure 4?)
* Table 1 and Table 2 both show BM > BL. (How do we square that with saying BL is the best?) Figure 5 also shows BM as best.

**Questions:**

Q: Were you able to validate the reasoning faithfulness or compare gpt-120's answers vs an expert? (Are the language model judgements grounded to something concrete?)
Q: "Best Line had more stable RL training than Best Move." If the Best Move dataset were scaled such that the total number of meaningful tokens of learning were the same, do we still see the improvement? (Which part of this approach is the key part?) (Is the centi pawn estimate the key part?)
Q: Having a place where it clearly defines what datasets are used when/where would help.  "Best Move - All"; Where is All defined? (I might have missed it.)

# Notes, Minor
* Overall the style and sizing of the barplot figures were hard to read. It did not jump out what was the metric and what was the training condition.
* Likewise, I didn't find the multi-part figures always helpful. Making them bigger with more stark lines would help.

---

> ### Author Response · Authors · 2025-11-21
> **Response**
>
> Appreciate the thorough review! Couple responses below:
>
> >Weaknesses 1 & 2: Language models so far aren't great at playing chess directly and trained model does not exceed/match the performance of the 120B model so it is unclear how well these findings scale. Or how meaningful they are.
> This work does not reconnect the training results with downstream/tournament/chess results. Further grounding the results w.r.t. to external models or performance would help position this work.
>
> This is a valid point — that said our intent and scope was to study how mid-training data influences downstream RL performance. So our goal was not necessarily to train the best chess model. Chess performance does serve as a core metric to understand model behavior from, but again our scope wasn’t as directly focused on direct chess playing ability — rather, how does data influence reasoning.
>
> It is worth noting that our model outperforms gpt-oss-120b (medium) on several tasks and outperforms Llama 4 Maverick (~50x the number of parameters) on all the tasks — even the OOD evaluations experiment we include in the general comment. We feel that these results add merit to our methods used -- though we'd like to reiterate that the goal of the paper was to study how data influences reasoning so much of our focus was on that (vs. how to make the best 'chess language model'). We mention this a bit in our limitations section and do note what we believe are techniques to further improve performance for those interested.
>
> >Weakness 5: Table 1 and Table 2 both show BM > BL. (How do we square that with saying BL is the best?) Figure 5 also shows BM as best.
>
> Good comment here that we're trying to tighten up in an updated version of the paper that we'll share because some of the results may seem contradicting.
>
> The reason we believe Best Line is better is the following:
> 1) It was more stable in training
> 2) It is more interpretable (reasoning quality perspective)
> 3) Arguably it has comparable performance to the Best Move dataset
> 4) It generalizes better to out of distribution data (see the general comment where we share the out of distribution data results)
>
> We made a change in the updated revision we just submitted that hopefully adds clarity here.
>
>
> >Weakness 3: I think the reasoning faithfulness metrics were not well justified (see questions).
> Q1: Were you able to validate the reasoning faithfulness or compare gpt-120's answers vs an expert? (Are the language model judgements grounded to something concrete?)
>
> Great points! This is something we conducted a lot of prompt tuning on and scrutinized heavily during training. However, we hadn't conducted a formal statistical analysis -- we decided to run this experiment and show our results in the general comment. For both experts we see these qualitative measurements are statistically significant along all dimensions.
>
> Couple things we'd like to note about these results:
> - Some of the reasoning can be really noisy so grading is tricky -- especially for poor performers or overly verbose / uninterpretable traces (e.g., gpt-oss-120b). We do feel confident, especially with the statistical results, that there is useful signal in these qualitative measurements.
> - Grading faithfulness is much easier -- "How well does the final move align with what reasoning was discussing?" It still is noisy, but as we see there is much higher agreement with experts here and we feel this measurement is validated as it underlies several of our larger takeaways.
>
>
> >Q2: "Best Line had more stable RL training than Best Move." If the Best Move dataset were scaled such that the total number of meaningful tokens of learning were the same, do we still see the improvement? (Which part of this approach is the key part?) (Is the centi pawn estimate the key part?)
>
> We're not entirely following this -- we would argue that both Best Move and Best Line are equal in the 'difficult token density' domain. There is likely an information theoretic way to show this, but if you consider the two tasks:
> - Best Move: 4 tokens per sample for each UCI move -> e.g. 'a2a4' means move piece from 'a2' to 'a4'
> - Best Line: ~25 tokens for a sample with 5 moves. You're predicting the best line -- we feel it is comparable difficulty when predicting each move in the optimal line compared to just predicting the best move. Then predicting the centipawn delta (e.g., '(\delta -25)') does have some structured tokens that become trivial to predict but as a whole these are a small percent of the tokens you train on.
>
> Regarding which tokens are most important for best line, our understanding from an early experiment was that the 'optimal line' tokens likely provide most of the quantitative performance boost. However this was a smaller scale run and we didn't test how the other aspects of reasoning changed. This could be an interesting experiment for further work.

---

> > ### Author Response · Authors · 2025-11-21
> > **Response (Cont.)**
> >
> > >Weakness 4: It took a whole to understand how/what data mixtures were used. Part of it is because Figure 4 really doesn't capture the information and "Best Line" and "Best Move" mixtures include other data as well. I think moving some of the information up from APP D DATA INCLUSION ANALYSES would help set this straight. Also, I think it is reasonable to do, but the final best run Best Move + Best Line also has 2x the data of other runs. (Which makes it unclear how you did the comparison for Figure 4?)
> > Q3: Having a place where it clearly defines what datasets are used when/where would help. "Best Move - All"; Where is All defined? (I might have missed it.)
> >
> > Definitely understand where this confusion can come from. We spent a lot of time deliberating on how we wanted to present our data -- we considered moving more of the methods up front (e.g., token distributions in App. D) but ultimately decided that we would move this information to the appendix so we can focus the 'core' of the paper on discussion of results and key takeaways. Being mindful of this tradeoff, we do try to make several references to where in the appendix someone can look to find this information.
> >
> > Regarding Figure 4 that is a valid point and ultimately came down to compute constraints. Ideally we would have a Best Move + Best Line experiment that was similarly 60mm tokens but Best Move data was costly to train. The purpose of Figure 4 was to highlight that including the Best Line data led to better training dynamics vs. our Best Move run.
> >
> > Lastly — we define the ‘All’ part of our data in a few places. In Appendix D we show the exact token distribution, but in the main part of the paper we mention this in the Section 4 introduction, Figure 5 caption, and Section 4.1. We definitely understand that following the exact datasets used can be confusing but we tried to be very clear and point to the appendix when possible.
> >
> > We spent a lot of time playing around with this and likely want to continue with our current format for the above mentioned reasons, but completely understand this comment.
> >
> > *Let us know if there are any further questions / thoughts regarding your provided score -- we'd be happy to address any other weaknesses further!*

---

### Official Review · Reviewer_JYqb · 2025-11-01

**Soundness:** 3
**Presentation:** 2
**Contribution:** 2
**Rating:** 4
**Confidence:** 3

**Summary:**

This paper investigates how "reasoning" ability emerges and evolves in a large language model as it undergoes supervised fine-tuning and subsequent reinforcement learning (RL). Using the game of chess as a verifiable, sequential decision-making environment (a Markov Decision Process), the authors meticulously study how different types of training data influence a model's performance and the nature of its internal logic.

**Strengths:**

1. The choice of chess is a strength of the paper. Chess is not just a game; it is a verifiable, discrete, episodic MDP, which provides a controlled environment for studying reasoning. The availability of chess engines as objective oracles for reward removes the need for potentially noisy human feedback or learned reward models, helping to isolate the variable of interest—the impact of training data on the reasoning process.
2. The discovery that SFT-checkpoint metrics can predict final RL performance is a practical contribution. The SFT stage is cheaper and faster than RL. By identifying these predictive signals, the authors provide a cost-effective methodology for iterating on and selecting the best base models before committing to expensive RL runs.

**Weaknesses:**

1. Limited Generalizability of the Domain: While a strength for verifiability, the sterile, perfect-information environment of chess is also a weakness. Real-world decision-making is noisy, partially observable, multi-agent, and often lacks a clear reward signal.
2. The paper lacks analysis regarding scalability.
3. Conflation of "Reasoning" with "Verbalization": The study defines reasoning as the model's language-based chain-of-thought. This is a limiting perspective. True reasoning might be occurring in a continuous latent space.

**Questions:**

Nil

---

> ### Author Response · Authors · 2025-11-21
> **Response**
>
> # Reviewer JYqb
>
> Thank you for reviewing -- please see below!
>
> >Weakness 1: Limited Generalizability of the Domain: While a strength for verifiability, the sterile, perfect-information environment of chess is also a weakness. Real-world decision-making is noisy, partially observable, multi-agent, and often lacks a clear reward signal.
>
> We are currently finalizing a few experiments generalizing this to other domains. Specifically we're doing this in AlfWorld and CookingWorld which are text-based domains that are tied to embodied AI tasks -- the intent here is to show that in broader sequential decision making domains these results hold.
>
>
> >Weakness 2: The paper lacks analysis regarding scalability.
>
> This would be interesting to further explore — though for this work we opted to focus our study on how midtraining data influences downstream RL performance. As a result we were less focused on explicit chess-playing ability and more on the various attributes of reasoning behavior and how they develop based on the data you train on.
>
> That said, we do show data scalability and show that training on more tokens (see scaled experiments) leads to much stronger performance than our initial inclusion analyses. Additionally, we run our experiments on a 7B parameter model -- we believe results on this size can be indicative of larger model peformance. Granted, further pursuing scalability on larger  models would be a valuable contribution, but this was out of scope and beyond our constraints.
>
>
> >Weakness 3: Conflation of "Reasoning" with "Verbalization": The study defines reasoning as the model's language-based chain-of-thought. This is a limiting perspective. True reasoning might be occurring in a continuous latent space.
>
> Yep! We fully agree with this interpretation and discuss this in the first paragraph of Section 2.1:
>
> ”Note that this reasoning need not be wholly interpretable – for example, it can exist in continuous space (Hao et al., 2024) or shift between languages (DeepSeek-AI et al., 2025) – what ultimately matters is that the intermediate steps are beneficial to the model. For this work we will focus on language-based reasoning.”
>
> There are many parts of our experiments that poke at this -- judging verbalized reasoning faithfulness does this. Additionally, our 'Factual Board Answering' dataset (an example is something like {Q: My piece on __ can take their bishop, A: b6}) was an attempt to see how latent board understanding translates to downstream reasoning. So definitely aligned that reasoning != verbalization and approach this in multiple parts of our work.
>
> *Given our responses here / to other reviewers — do you have any other questions or thoughts that motivates your provided rating of a 4? Would be happy to answer questions or include more detail in the final version!*

---

### Official Review · Reviewer_GB2Q · 2025-11-01

**Soundness:** 3
**Presentation:** 4
**Contribution:** 2
**Rating:** 6
**Confidence:** 4

**Summary:**

In the controlled domain of chess, this study dives into which types of tasks/rewards yield best performance in terms of accuracy and reasoning faithfulness during supervised finetuning (SFT). It finds that while move prediction alone is effective, it is unfaithful and that a mixed multi-turn trajectory (with other board reasoning tasks) leads to similar performance with increased faithfulness. When RL is further applied on top of these checkpoints, the latter (more faithful) also leads to stabler RL training and better performance. This study then concludes that the metrics of SFT checkpoints can correlate with the eventual effectiveness of the RL.

**Strengths:**

* The study is well-motivated and self-contained. The study thoroughly investigates the hypotheses around how SFT datasets impact both SFT and RL performance and how qualitative performance and downstream performance are affected by the SFT data.
* The analysis is comprehensive across several different data mixes for SFT and chess board reasoning tasks, and the findings to the research questions are interesting and thought-provoking, even if the actual performances are not strong or the differences in final performance are not significant. The paper is still well-organized despite all the research questions, SFT methods/models, and experiments.

**Weaknesses:**

The focused study of the chess domain (which is omitted from the title) makes it difficult to draw strong conclusions to other domains like language (or multimodal) reasoning tasks. In particular, there’s a less natural notion of (best) “next move” for reasoning problems - the basic assumption for reasoning problems is already focused on the line (full trajectory). This is less of an issue with some of the other board reasoning tasks, but there is perhaps too much focus on best move/line. Even though they are most effective, they may be less relatable to problems in other domains.

More concretely, there is no parallel “application” of these learnings to real-world datasets or reasoning problems to verify these findings.

Ultimately, I view this as a major weakness, moreso than some of the other limitations described in the paper like model size or over-focus on specific aspects of chess.

**Questions:**

1. Why didn’t the board understanding questions and verbalized alpha-beta pruning work? I don't completely understand the token density argument because many reasoning LLMs prioritize longer response length as a positive signal for "reasoning" capabilities. So what's different in chess? Does it imply that an optimal model (like chess engines) would not be able to reason about chess particularly well outside of giving good moves/board valuations, or is it more likely a failure of the instruction tuning dataset mix and that it could be fixed? There’s a related observation where the unfaithful reasoning still yielded fair performance, so perhaps the model doesn't need to understand why a move is good to think it is better.


2. Relatedly, the final evaluation are the tasks described in 3.2, which are geared for the game of chess itself. So it is perhaps unsurprising that while a mix is helpful, line and best move are most critical (and therefore qualitative signals like move accuracy/reasoning quality correlate). But how do these do on the other tasks in the dataset, like board factual QA (the other two tasks are likely harder to evaluate). Would your conclusions be starkly different? Is the model too overfit to think only about moves?

3. Even though the best move and best line SFT modes yielded similar performances, did they differ in their confidences of predictions?

---

> ### Author Response · Authors · 2025-11-21
> **Response**
>
> Thank you for your review! We wanted to address each part separately -- see below.
>
> >Weakness: Generalization to other domains.
>
> We are currently finalizing a few experiments generalizing this to other domains. Specifically we're doing this in AlfWorld and CookingWorld which are text-based domains that are tied to embodied AI tasks -- the intent here is to show that in broader sequential decision making domains these results hold. We'll upload these results ASAP.
>
>
> >Q1: Why didn’t the board understanding questions and verbalized alpha-beta pruning work? I don't completely understand the token density argument because many reasoning LLMs prioritize longer response length as a positive signal for "reasoning" capabilities. So what's different in chess? Does it imply that an optimal model (like chess engines) would not be able to reason about chess particularly well outside of giving good moves/board valuations, or is it more likely a failure of the instruction tuning dataset mix and that it could be fixed? There’s a related observation where the unfaithful reasoning still yielded fair performance, so perhaps the model doesn't need to understand why a move is good to think it is better.
>
> Fair questions! There are definitely some unintuitive results from these experiments -- many of which have created ideas for future work as reasoning is odd :) Splitting this question up since there are a couple parts:
> - *Why didn't factual board understanding (FBA) / verbalized alpha-beta pruning (VABP) work?* Here are some of our thoughts on why these were less effective.
>     - VABP: We think the key failure comes from two things.
>         - First, the 'difficult token density' argument. In our synthetic data generation algorithm we built phrase banks (e.g., random_sample(‘We could respond with ___’, ‘We could also play ___’, …) ) with the intent to make explanations sound like natural language. However, these phrases (order of ~200 total) are memorizable; given we trained on a fixed token budget, these phrases were quickly memorized and became trivial to predict (low entropy, negligible gradient updates). Then the density of interesting, non-trivial tokens (moves, valuations, branching decisions) were at a lower density (possibly one in every 10 tokens) which meant this dataset 'taught' the model less than other datasets.
>         - Second, asking a language model to verbalize tree search is an inherently difficult task. It requires structured, algorithmic thinking that can be difficult for language models to reliably do -- especially smaller models (like we used). In addition to this, the model would need to effectively choose the best move branches to search as they are limited in the number of moves they can realistically explore; chess engines may search O(1M+) nodes in making a move decision, language models are likely limited to O(10). One final point is that proper tree search like this with backtracking requires 'arbitrary state reset' abilities which language models currently struggle with (e.g., 'context rot' but also general performance degradation on evals as context length grows).
>     - FBA: This is a more surprising result as the 'non-trivial token density' argument doesn't apply here. In fact we designed this dataset specifically to address that issue and to see if latent board understanding translated to downstream reasoning behavior. But before sharing our speculation, we want to highlight what we can gather about this particular model.
>         - **What experiments showed:** (A) First, the initial base model is essentially guessing. If you consider the 'Is Legal' task -- this asks for a response of {Yes, No} and our base model got 49% accuracy. Do note this task was designed to be tricky for LLMs with many 'gotchas' -- and all 'illegal' moves would be legal if the player wasn't in check or there wasn't a piece in the way. (B) Second, when we include this FBA data in our datamix, we begin to see non-trivial performance. Clearly the model is learning to do this task: During SFT the validation loss drops below 0.1 for unseen data in the FBA task (equates to ~0.9 probability given to the correct token on average); the evaluations we uploaded above also similarly show the model learns this task non-trivially. So combining these two points, **There are useful general chess capabilities we bake into the model (B) that didn't exist in the base (A).**
>         - **Why doesn't this lead to better chess playing models?** This is speculation -- it is something we may be pursuing for our next work as understanding and improving this has implications on broad generalization :) The naive interpretation is that the downstream task (play good chess move) is just too different; the prompt and question (model context) differs too heavily from the FBA tasks that the model doesn't end up 'leveraging' this new latent ability. Definitely unintuitive results, we candidly expected more positive transfer here.

---

> > ### Author Response · Authors · 2025-11-21
> > **Response (cont.)**
> >
> > - *I don't completely understand the token density argument because many reasoning LLMs prioritize longer response length as a positive signal for "reasoning" capabilities. So what's different in chess?*
> >     - We hope the 'VABP' sub-bullet above addressed the 'difficult token density' argument. Regarding LLMs prioritizing longer responses, there is a lot of nuance here. Some of the 'longer response' results we initially saw from DeepSeek R1 was due to a quirk of naive GRPO -- see Dr. GRPO (Liu 2025, Understanding R1-Zero-Like Training: A Critical Perspective) that talks about this. But more recent work (Khatri 2025, The Art of Scaling Reinforcement Learning Compute for LLMs) shows that increasing your context length does slightly increase your reward asymptote (thought takes much longer to train). So the jury is still out; generally more tokens = better performance but that isn't direct causation as the reasoning needs to be a certain quality to be beneficial. You can easily run into instability where your model begins repeating itself as a counterpoint.
> > - *Does it imply that an optimal model (like chess engines) would not be able to reason about chess particularly well outside of giving good moves/board valuations, or is it more likely a failure of the instruction tuning dataset mix and that it could be fixed? There’s a related observation where the unfaithful reasoning still yielded fair performance, so perhaps the model doesn't need to understand why a move is good to think it is better.*
> >     - We don't think that would be the implication. If you had an LLM that was 'superhuman' at chess and 'reasoned' with a form of alpha-beta pruning, it would likely have the ability to do various chess-related tasks within its latents (assuming those tasks are useful for playing strong chess). However, how to extract this ability -- whether through prompting, probing, fine-tuning -- is a different question. At least that is our belief -- so it could probably be fixed through the right data. Regarding "is reasoning necessary to actually play chess", that is a valid question! The DeepMind chess transformer paper (Ruoss 2024, Grandmaster-Level Chess Without Search) showed that purely neural-based methods and no 'reasoning' can lead to grandmaster level. Note they tested a direct policy (predict move given board) and this was less effective but still very strong -- their grandmaster level was using a action-value function (like Q-learning). Ideally a language model could do both -- play a good move and know why it is a good move -- but as we see reasoning is odd!
> >
> >
> > >Q2: Relatedly, the final evaluation are the tasks described in 3.2, which are geared for the game of chess itself. So it is perhaps unsurprising that while a mix is helpful, line and best move are most critical (and therefore qualitative signals like move accuracy/reasoning quality correlate). But how do these do on the other tasks in the dataset, like board factual QA (the other two tasks are likely harder to evaluate). Would your conclusions be starkly different? Is the model too overfit to think only about moves?
> >
> > Great questions -- we went ahead and ran some experiments on all our models that we share in a general comment. If you consider our inclusion experiments that did not train on FBA tasks -- particularly the 'Best Move' and 'Best Line' ones -- we see that actually performance on the board answering task is no different from the other datasets.
> >
> > So being able to play better moves (which these two models do -- see Figure 5/App. D) doesn't lead to better board understanding. As we discussed above, vice-versa this is also the case. So minimal positive transfer in either direction.
> >
> > >Q3: Even though the best move and best line SFT modes yielded similar performances, did they differ in their confidences of predictions?
> >
> > This is an interesting idea that we haven’t considered exploring — there are a few proxies you could use to estimate this but there isn’t a perfect measurement of uncertainty as the full reasoning trace likely sharpens the final prediction (so something like entropy of the first token in the answer tags may not tell the full picture).
> >
> > Due to time constraints we don’t believe we’ll be able to implement this test but it is an consideration for future work.
> >
> > *Would appreciate hearing if there are any other thoughts / questions / weaknesses you have -- happy to address!*

---

> > > ### Comment · Reviewer_GB2Q · 2025-11-28
> > >
> > > Thank you for your thoughtful answers, there definitely seems to be something unexpected and interesting here around positive transfer between board understanding and winning -- and maybe there are more general hypotheses that could be drawn for problem solving or tool use tasks.
> > >
> > > At the moment, my score remains unchanged.

---

### Author Response · Authors · 2025-11-21
**Paper Revisions (Log)**

(11.20.2025)
- Added statistical analysis of expert vs. LLM grading to Appendix G
- Created Appendix I: Reinforcement Learning Training Dynamics
- Refreshed data in Figures 6 and 7 as we found a bug that discarded some of the qualitative analysis samples in the initial numbers. We do not feel this resulted in any changes to our key results. Note that the only change we made was w.r.t. inference parameters (specifically sequence length) -- no changes were made to prompting, temperature, etc. that could materially adjust scoring. Flagging that after rerunning this analysis, the 'Guided Synthetic' results that we previously made a small note of were no longer as significant to results so this was removed from Figure 6.
- Inclusion of our new evaluation (tasks not trained on during the RL stage) in Appendix D -- we add both results and example problems. Additionally, we briefly reference these results at the end of section 4.1 when comparing the performance of the Best Move and Best Line experiments, flagging that the Best Line experiments were more robust on these OOD evaluations.

---

### Author Response · Authors · 2025-11-21
**New Experimental Results (Statistical Analysis - LLM-Judge vs. Expert)**

## Statistical Analysis of Expert vs. LLM Grading (Qualitative Metrics)
We formalize that LLM judge score correlation with experts is statistically significant. Our LLM-as-a-judge prompting was tuned to align with experts initially but we hadn't formally executed as a statistical analysis.

Samples are a random sample from a set containing outputs from all tested models (both SFT and RL checkpoints). gpt-oss-120b responses are excluded as LLM judges have shown a tendency to favor outputs from itself (Panickssery 2024, LLM Evaluators Recognize and Favor Their Own Generations).

| Expert   | N  | Efficacy       |             | Efficiency     |             | Faithfulness   |             | Sum of All |             |
|----------|----|----------------|-------------|----------------|-------------|----------------|-------------|----------------|-------------|
|          |    | Corr           | P-value     | Corr           | P-value     | Corr           | P-value     | Corr           | P-value     |
| Expert 1 | 96 | 0.41           | 4.01e-05    | 0.31           | 2.03e-03    | 0.68           | 1.50e-14    | 0.57           | 1.10e-09    |
| Expert 2 | 73 | 0.24           | 4.00e-02    | 0.34           | 3.07e-03    | 0.43           | 1.71e-04    | 0.39           | 6.63e-04    |

We have included this analysis in Appendix G (Reasoning Quality) of our paper.

---

### Author Response · Authors · 2025-11-21
**New Experimental Results (OOD Evaluation Performance)**

## Evaluation of Models on Other Tasks (OOD)
We tested models on two other tasks to understand how well they perform in alternate, out-of-distribution tasks:
1) FBA: How well do they perform on unseen 'Factual Board Answering' questions? This is a score over 1,000 questions across 5 separate tasks. Each task has a score w/in [0, 1] -- so perfect would be 1.00.
    - Do note that 'Factual Board Answering', 'Best Move - All', 'Best Line - All', 'Best Move XL', 'Best Line XL', and 'Best move + Best Line XL' have FBA data in their training sets. That said, all tests are on unseen tasks and these tasks are asked to be provided within answer tokens -- the training data does SFT directly on the answers (no reasoning or answer tags). Because of this, 'Best Move - All' significantly underperforms as it gives improper parsing formatting on a few of the tasks.
2) OOD Mates: This is a test where on 3 separate out of distribution domains, the model is asked to correctly choose a checkmate given the current position. These domains are 'Knights & Rooks', 'Same Color', and 'More Pieces' from *Mészáros 2025, Out-of-distribution Tests Reveal Compositionality in Chess Transformers*. For example, Knights & Rooks has a massive number of knights and rooks on the board and the goal is to checkmate the opposing king -- these 'Knights & Rooks' board states are impossible to reach in normal play. The others are very rare to reach but possible given various abnormal pawn promotions.

| Model                           | FBA       |           | OOD Mates |           |
|---------------------------------|-----------|-----------|-----------|-----------|
| **Baselines**                   |           |           |           |           |
| Qwen2.5 7B-Instruct             | 0.32      | —         | 0.00      | —         |
| Llama 4 Maverick                | 0.47      | —         | 0.14      | —         |
| gpt-oss-120b (Medium)           | **1.00**  | —         | **0.78**  | —         |
|                                 |           |           |           |           |
| **Inclusion Experiments**       | *SFT*     | *RL*      | *SFT*     | *RL*      |
| Rejection Sampling - Pred. Move | 0.38      | 0.37      | 0.04      | 0.01      |
| Rejection Sampling - All        | 0.41      | 0.32      | 0.04      | 0.02      |
| Verbalized Alpha-Beta Pruning   | 0.41      | 0.34      | 0.04      | 0.01      |
| Factual Board Answering         | **0.60**  | **0.58**  | 0.06      | 0.04      |
| Guided Synthetic                | 0.42      | 0.36      | 0.04      | 0.02      |
| Best Move                       | 0.36      | 0.34      | 0.07      | 0.06      |
| Best Line                       | 0.40      | 0.37      | 0.06      | 0.08      |
| Best Move - All                 | 0.37      | 0.37      | 0.06      | 0.05      |
| Best Line - All                 | 0.57      | **0.58**  | **0.09**  | **0.14**  |
|                                 |           |           |           |           |
| **Scaled Experiments**          | *SFT*     | *RL*      | *SFT*     | *RL*      |
| Best Move XL                    | 0.40      | 0.71      | 0.07      | 0.05      |
| Best Line XL                    | **0.61**  | 0.65      | 0.08      | 0.09      |
| Best Move + Best Line XL        | 0.59      | **0.82**  | **0.11**  | **0.15**  |

A few things we'd like to note on common failure nodes:
- For the FBA task, we often see the model predict the correct answer immediately (similar to the SFT task) then proceed to enter its reasoning and provide a final answer in parsed answer tags. Often this first prediction is correct when the final answer is incorrect. As these are all new unseen tasks, this is likely a sign that the model's latent abilities that are used when predicting the next token directly aren't fully leveraged during its final reasoning, which partially aligns with our finding about unfaithful reasoning for the predict move task.
- For the OOD Mates, our best performance is candidly lower than we would have hoped. That said it is a significant improvement on the 0% of the base model and beats Llama 4 Maverick (that has ~50x the parameter count of our model). These are quite out of domain examples and some take time to determine the right answer as they aren't immediately obvious -- we believe that if one employed some of the strategies we mention in our limitations section (multi-turn RL, tuning the reward for Predict Move), you could strongly improve this OOD generalization.

We have added this to our latest paper revision and we have examples of the OOD Mate problems as well (See Appendix D).

---

### Author Response · Authors · 2025-12-03
**Summary of Key Changes & Discussion Period**

Hello! We'd like to provide a summary of the discussion period to allow a new AC to get up to speed quickly.
- Reviewers 1Npe and sfa6 asked if there was a statistical analysis on our LLM-judge scores vs. human experts. We posted the results (and added to the paper -- Appx. G) of blind grading vs. two experts which showed statistically significant correlation on all metrics.
- Reviewers GB2Q and 1Npe had comments that led us to conduct an additional evaluation of out-of-domain tasks. We added these results to the paper (Appx. D) and incorporated these results into some commentary on robustness of the different final checkpoints.
- We included reinforcement learning training dynamics into the paper (Appx. I) per a comment from reviewer sfa6.
- We also directly responded to several one-off questions about our results and findings that reviewers had -- these are included in the below discussion comments.

We would also like to provide an update on domain extension work that we mentioned in two comments. We are seeing positive initial results in Textworld (Cote, 2018) environments but believe domain extension is beyond the scope of this current paper for the following reasons:
- A true generalization study should include additional environments -- we believe solid candidates would include coding, dialogue, and text-based embodied games (e.g., ALFworld, Shridhar 2020). For context, there has been some recent work in coding (CodeI/O, Li 2025)(Absolute Zero, Zhao 2025) that has shown that training a model to predict outputs from code + code inputs leads to improved performance across many tasks
- The intention of this paper was to study very controlled interventions in a well-behaved self-contained environment (chess). We think this work has clear contributions to the broader research community as it stands given we address our motivating questions and examine how reasoning evolves from midtraining data from many angles, a problem we believe is understudied and that we do address

We've added to the discussion section to highlight what we feel is promising future work on how training a language model on environment dynamics can influence your reasoning to signal this domain extension direction.

---

### Meta-Review · Area_Chair_RVCR · 2026-01-07

**Summary:**

This paper studies how reasoning behaviour in language models evolves under different post-training data mixtures and training regimes (SFT and RL) in a sequential decision-making domain, using chess as a controlled, verifiable testbed. Reviewers agree the topic is interesting and that the empirical effort is substantial. However, the overall assessment trends toward reject due to concerns about generalizability beyond chess, clarity and rigor of experimental presentation, and strength of the claimed insights relative to the narrowness of the domain. While the rebuttal was thorough and added meaningful analyses (OOD evaluation, statistical validation of LLM judges, and RL training dynamics), they don't do much to address the concerns that the findings don't seem to support strong, broadly applicable conclusions about reasoning evolution in language models.

**Reviewer Concerns:**

Concerns fully or mostly addressed by the rebuttal:
* Validation of reasoning quality metrics: The authors added a statistical comparison between LLM-as-a-judge scores and human expert evaluations, showing significant correlations across metrics. This directly addressed concerns raised by multiple reviewers about the grounding of qualitative reasoning and faithfulness scores.
* OOD robustness: Additional evaluations on OOD tasks (Factual Board Answering and OOD checkmate problems) partially addressed concerns about overfitting to in-domain chess evaluations and helped clarify differences between training regimes.
* RL training dynamics: The inclusion of RL training dynamics (response length, stability trends) addressed requests for more mechanistic insight into why certain SFT checkpoints lead to more stable or effective RL.
* Clarity around unexpected negative results: Reviewers’ questions about why certain datasets (eg, verbalized alpha-beta pruning, board understanding) underperformed were answered with detailed analysis and plausible hypotheses.

Concerns that remain outstanding
* Limited generalizability: Multiple reviewers emphasized that conclusions drawn from chess (deterministic, perfect-information, with a clear oracle) might not transfer to broader reasoning domains such as math, coding, or natural language problem-solving. While domain extension was discussed and preliminary results were mentioned, convincing cross-domain evidence is not yet present. This is problematic given the generality implied by the title and abstract.
* Strength of contribution: Some reviewers viewed the work as primarily an extensive empirical case study rather than a contribution that yields new general principles or methods for reasoning models. The absence of a concrete algorithmic or modeling advance weakened the perceived impact.
* Experimental clarity and presentation: Several reviewers noted that figures and tables were difficult to interpret (e.g., dense or small plots, confusing data mixtures, unclear comparisons when training budgets differed). Although explanations were provided in responses and appendices, these issues remain visible in the paper itself, even now.
* Scaling and competitiveness: Reviewers questioned how meaningful the findings are given that the trained models do not approach the performance of much larger or stronger baselines, leaving uncertainty about how the observed effects scale with model size or strength.

**Reviewer Scores:**

* Reviewer GB2Q: Likely remains at 6. While acknowledging interesting findings and appreciating the rebuttal, this reviewer maintained that the lack of broader applicability beyond chess is a major weakness and explicitly stated their score would not change.
* Reviewer 1Npe: Likely remains at 4.  The rebuttal analyses addressed some concerns (faithfulness validation, OOD tests), but issues around clarity of comparisons, scaling, and presentation likely persist.
* Reviewer sfa6: Likely remains at 6 since they don't seem overly enthusiastic about the paper to champion it, and many of their concerns about generality remain unaddressed.
* Reviewer JYqb: Likely remains at 4. Even if some concerns were addressed, the reviewer’s skepticism about generalizability and scalability likely keeps the score at 4. I've partially discounted this review in any case given its brevity, AI-generation telltale signs, and the reviewer's expertise being unrelated to this paper.

Overall: After accounting for likely score adjustments, the paper would still sit in borderline territory with no clear consensus in favor. Given the remaining (and quite important) concerns about generality, presentation quality, and strength of contribution, I recommend reject.

---

### Decision · Program_Chairs · 2026-01-26

Reject